# Analyzing and Improving Greedy 2-Coordinate Updates for Equality-Constrained Optimization via Steepest Descent in the 1-Norm.

## Abstract

We consider minimizing a smooth function subject to a summation constraint over its variables. By exploiting a connection between the greedy 2-coordinate update for this problem and equality-constrained steepest descent in the 1-norm, we give a convergence rate for greedy selection under a proximal Polyak-Łojasiewicz assumption that is faster than random selection and independent of the problem dimension $n$. We then consider minimizing with both a summation constraint and bound constraints, as arises in the support vector machine dual problem. Existing greedy rules for this setting either guarantee trivial progress only or require $O(n^2)$ time to compute. We show that bound- and summation-constrained steepest descent in the L1-norm guarantees more progress per iteration than previous rules and can be computed in only $O(n \log n)$ time.

## 1 Introduction

Coordinate descent (CD) is an iterative optimization algorithm that performs a gradient descent step on a single variable at each iteration. CD methods are appealing because they have a convergence rate similar to gradient descent, but for some common objective functions the iterations have a much lower cost. Thus, there is substantial interest in using CD for training machine learning models.

**Unconstrained coordinate descent**: Nesterov [2012] considered CD with random choices of the coordinate to update, and proved non-asymptotic linear convergence rates for strongly-convex functions with Lipschitz-continuous gradients. It was later shown that these linear convergence rates are achieved under a generalization of strong convexity called the Polyak-Łojsiewicz condition [Karimi et al., 2016]. Moreover, greedy selection of the coordinate to update also leads to faster rates than random selection [Nutini et al., 2015]. These faster rates do not depend directly on the dimensionality of the problem due to an equivalence between the greedy coordinate update and steepest descent on all coordinates in the 1-norm. For a discussion of many other problems where we can implement greedy selection rules at similar cost to random rules, see Nutini et al. [2022, Sections 2.4-2.5].

**Bound-constrained coordinate descent**: CD is commonly used for optimization with lower and/or upper bounds on each variable. Nesterov [2012] showed that the unconstrained rates of randomized CD can be achieved under these separable constraints using a projected-gradient update of the coordinate. Richtárik and Takáč [2014] generalize this result to include a non-smooth but separable term in the objective function via a proximal-gradient update; this justifies using CD in various constrained and non-smooth settings, including least squares regularized by the 1-norm and support vector machines with regularized bias. Similar to the unconstrained case, Karimireddy et al. [2019] show that several forms of greedy coordinate selection lead to faster convergence rates than random selection for problems with bound constraints or separable non-smooth terms.

**Equality-constrained coordinate descent**: many problems in machine learning require us to satisfy an equality constraint. The most common example is that discrete probabilities must sum to one. Another common example is SVMs with an unregularized bias term. The (non-separable) equality constraint cannot be maintained by single-coordinate updates, but it can be maintained if we update two variables at each iteration. Necoara et al. [2011] analyze random selection of the two coordinates to update, while Fang et al. [2018] discuss randomized selection with tighter rates. The LIBSVM package [Chang and Lin, 2011] uses a greedy 2-coordinate update for fitting SVMs which has the same cost as random selection. But despite LIBSVM being perhaps the most widely-used CD method of all time, current analyses of greedy 2-coordinate updates either result in sublinear convergence rates or do not lead to faster rates than random selection [Tseng and Yun, 2009, Beck, 2014].

**Our contributions**: we first give a new analysis for the greedy 2-coordinate update for optimizing a smooth function with an equality constraint. The analysis is based on an equivalence between the greedy update and equality-constrained steepest descent in the 1-norm. This leads to a simple dimension-independent analysis of greedy selection showing that it can converge substantially faster than random selection. Next, we consider greedy rules when we have an equality constraint and bound constraints. We argue that the rules used by LIBSVM cannot guarantee non-trivial progress on each step. We analyze a classic greedy rule based on maximizing progress, but this analysis is dimension-dependent and the cost of implementing this rule is $O(n^2)$ if we have both lower and upper bounds. Finally, we show that steepest descent in the 1-norm with equalities and bounds guarantees a fast dimension-independent rate and can be implemented in $O(n \log n)$. This rule may require updating more than 2 variables, in which case the additional variables can only be moved to their bounds, but this can only happen for a finite number of early iterations.

## 2 Equality-Constrained Greedy 2-Coordinate Updates

We first consider the problem of minimizing a twice-differentiable function $f$ subject to a simple linear equality constraint,

$$\min_{x \in \mathbb{R}^n} f(x), \quad \text{subject to} \sum_{i=1}^{n} x_i = \gamma, \tag{1}$$

where $n$ is the number of variables and $\gamma$ is a constant. On iteration $k$ the 2-coordinate optimization method chooses a coordinate $i_k$ and a coordinate $j_k$ and updates these two coordinates using

$$x_{i_k}^{k+1} = x_{i_k}^k + \delta^k, \quad x_{j_k}^{k+1} = x_{j_k}^k - \delta^k, \tag{2}$$

for a scalar $\delta^k$ (the other coordinates are unchanged). We can write this update for all coordinates as

$$x^{k+1} = x^k + d^k, \quad \text{where } d_{i_k}^k = \delta^k, d_{j_k}^k = -\delta^k, \text{ and } d_m^k = 0 \text{ for } m \notin \{i_k, j_j\}. \tag{3}$$

If the iterate $x^k$ satisfies the constraint then this update maintains the constraint. In the coordinate gradient descent variant of this update we choose $\delta^k = -\frac{\alpha^k}{2}(\nabla_{i_k} f(x^k) - \nabla_{j_k} f(x^k))$ for a step size $\alpha_k$. This results in an update to $i_k$ and $j_k$ of the form

$$x_{i_k}^{k+1} = x_{i_k}^k - \frac{\alpha^k}{2}(\nabla_{i_k} f(x^k) - \nabla_{j_k} f(x^k)), \ x_{j_k}^{k+1} = x_{j_k}^k - \frac{\alpha^k}{2}(\nabla_{j_k} f(x^k) - \nabla_{i_k} f(x^k)). \tag{4}$$

If $f$ is continuous, this update is guaranteed to decrease $f$ for sufficiently small $\alpha^k$. The greedy rule chooses the coordinates to update by maximizing the difference in their partial derivatives,

$$i_k \in \arg\max_i \nabla_i f(x^k), \quad j_k \in \arg\min_j \nabla_j f(x^k). \tag{5}$$

At the solution of the problem we must have partial derivatives being equal, so intuitively this greedy choice updates the coordinates that are furthest above/below the average partial derivative. This choice also minimizes the set of 2-coordinate quadratic approximations to the function (see Appendix A.1)

$$\arg\min_{i,j} \left\{ \min_{d_{ij} | d_i + d_j = 0} f(x^k) + \nabla_{ij} f(x^k)^T d_{ij} + \frac{1}{2\alpha^k} \|d_{ij}\|^2 \right\}, \tag{6}$$

which is a special case of the Gauss-Southwell-q (GS-q) rule of Tseng and Yun [2009].

We assume that the gradient of $f$ is Lipschitz continuous, and our analysis will depend on a quantity we call $L_2$. The quantity $L_2$ bounds the change in the 2-norm of the gradient with respect to any two coordinates $i$ and $j$ under a two-coordinate update of any $x$ of the form (3).

$$\|\nabla_{ij}f(x+d) - \nabla_{ij}f(x)\|_2 \leq L_2\|d\|_2. \tag{7}$$

Note that $L_2$ is less than or equal to the Lipschitz constant of the gradient of $f$.

## 2.1 Connections between Greedy 2-Coordinate Updates and the 1-Norm

Our analysis relies on several connections between the greedy update and steepest descent in the 1-norm, which we outline in this section. First, we note that vectors $d^k$ of the form (3) satisfy $\|d^k\|_1^2 = 2\|d^k\|_2^2$, since

$$
\begin{aligned}
\|d^k\|_1^2 &= (|\delta^k| + |-\delta^k|)^2 \\
&= (\delta^k)^2 + (\delta^k)^2 + 2|\delta^k| \cdot |\delta^k| \\
&= 4(\delta^k)^2 \\
&= 2((\delta^k)^2 + (-\delta^k)^2) \\
&= 2\|d^k\|_2^2.
\end{aligned}
$$

Second, if a twice-differentiable function's gradient satisfies the 2-coordinate Lipschitz continuity assumption (7) with constant $L_2$, then the full gradient is Lipschitz continuous in the 1-norm with constant $L_1 = L_2/2$ (see Appendix B). Finally, we note that applying the 2-coordinate update (4) is an instance of applying steepest descent over all coordinates in the 1-norm. In particular, in Appendix A.2 we show that steepest descent in the 1-norm always admits a greedy 2-coordinate update as a solution.

**Lemma 2.1.** *Let $\alpha > 0$. Then at least one steepest descent direction with respect to the 1-norm has exactly two non-zero coordinates. That is,*

$$\min_{d \in \mathbb{R}^n | d^T 1 = 0} \nabla f(x)^T d + \frac{1}{2\alpha}\|d\|_1^2 = \min_{i,j}\left\{ \min_{d_{ij} \in \mathbb{R}^2 | d_i + d_j = 0} \nabla_{ij} f(x)^T d_{ij} + \frac{1}{2\alpha}\|d_{ij}\|_1^2 \right\}. \tag{8}$$

This lemma allows us to equate the progress of greedy 2-coordinate updates to the progress made by a full-coordinate steepest descent step descent step in the 1-norm.

## 2.2 Proximal-PL Inequality in the 1-Norm

For lower bounding sub-optimality in terms of the 1-norm, we introduce the proximal-PL inequality in the 1-norm. The proximal-PL condition was introduced to allow simpler proofs for various constrained and non-smooth optimization problems [Karimi et al., 2016]. The proximal-PL condition is normally defined based on the 2-norm, but we define a variant for the summation-constrained problem where distances are measured in the 1-norm.

**Definition 2.2.** A function $f$, that is $L_1$-Lipschitz with respect to the 1-norm and has a summmation constraint on its parameters, satisfies the proximal-PL condition in the 1-norm if for a positive constants $\mu_1$ we have

$$\frac{1}{2}\mathcal{D}(x, L_1) \geq \mu_1(f(x) - f^*), \tag{9}$$

for all $x$ satisfying the equality constraint. Here, $f^*$ is the constrained optimal function value and

$$\mathcal{D}(x, L) = -2L \min_{\{y \ | y^T 1 = \gamma\}}\left[ \langle \nabla f(x), y - x \rangle + \frac{L}{2}\|y - x\|_1^2 \right]. \tag{10}$$

It follows from the equivalence between norms that summation-constrained functions satisfying the proximal-PL condition in the 2-norm will also satisfy the above proximal-PL condition in the 1-norm. In particular, if $\mu_2$ is the proximal-PL constant in the 2-norm, then we have $\frac{\mu_2}{n} \leq \mu_1 \leq \mu_2$ (see Appendix C). Functions satisfying these conditions include any strongly-convex function $f$ as well as relaxations of strong convexity, such as functions of the form $f = g(Ax)$ for a strongly-convex $g$ and a matrix $A$ [Karimi et al., 2016]. In the $g(Ax)$ case $f$ is not strongly-convex if $A$ is singular, and we note that the SVM dual problem can be written in the form $g(Ax)$.

## 2.3 Convergence Rate of Greedy 2-Coordinate Updates under Proximal-PL

We analyze the greedy 2-coordinate method under the proximal-PL condition based on the connections to steepest descent in the 1-norm.

**Theorem 2.3.** *Let $f$ be a twice-differentiable function whose gradient is 2-coordinate-wise Lipschitz (7) and restricted to the set where $x^T 1 = \gamma$. If this function satisfies the proximal-PL inequality in the 1-norm (9) for some positive $\mu_1$, then the iterations of the 2-coordinate update (4) with $\alpha^k = 1/L_2$ and the greedy rule (5) satisfy:*

$$f(x^k) - f(x^*) \leq \left(1 - \frac{2\mu_1}{L_2}\right)^k (f(x^0) - f^*). \tag{11}$$

*Proof.* Starting from the descent lemma restricted to the coordinates $i_k$ and $j_k$ we have

$$f(x^{k+1}) \leq f(x^k) + \nabla_{i_k j_k} f(x^k)^T d_{i_k j_k} + \frac{L_2}{2} \|d_{i_k j_k}\|^2$$

$$= f(x^k) + \min_{i,j} \left\{ \min_{\substack{d_{ij} \in \mathbb{R}^2 | \\ d_i + d_j = 0}} \nabla_{ij} f(x^k)^T d_{ij} + \frac{L_2}{2} \|d_{ij}\|^2 \right\} \quad \text{(GS-q rule)}$$

$$= f(x^k) + \min_{i,j} \left\{ \min_{\substack{d_{ij} \in \mathbb{R}^2 | \\ d_i + d_j = 0}} \nabla_{ij} f(x^k)^T d_{ij} + \frac{L_2}{4} \|d_{ij}\|_1^2 \right\} \quad (\|d\|_1^2 = 2\|d\|^2)$$

$$= f(x^k) + \min_{i,j} \left\{ \min_{\substack{d_{ij} \in \mathbb{R}^2 | \\ d_i + d_j = 0}} \nabla_{ij} f(x^k)^T d_{ij} + \frac{L_1}{2} \|d_{ij}\|_1^2 \right\} \quad (L_1 = L_2/2)$$

$$= f(x^k) + \min_{d | d^T 1 = 0} \left\{ \nabla f(x^k)^T d + \frac{L_1}{2} \|d\|_1^2 \right\} \quad \text{(Lemma 2.1)}.$$

Now subtracting $f^*$ from both sides and using the definition of $\mathcal{D}$ from the proximal-PL assumption,

$$f(x^{k+1}) - f(x^*) \leq f(x^k) - f(x^*) - \frac{1}{2L_1} \mathcal{D}(x^k, L_1)$$

$$= f(x^k) - f(x^*) - \frac{\mu_1}{L_1} (f(x^k) - f^*)$$

$$= f(x^k) - f(x^*) - \frac{2\mu_1}{L_2} (f(x^k) - f^*)$$

$$= \left(1 - \frac{2\mu_1}{L_2}\right) (f(x^k) - f^*)$$

Applying the inequality recursively completes the proof. $\square$

Note that the above rate also holds if we choose $\alpha^k$ to maximally decrease $f$, and the same rate holds up to a constant if we use a backtracking line search to set $\alpha^k$.

## 2.4 Comparison to Randomized Selection

If we sample the two coordinates $i_k$ and $j_k$ from a uniform distribution, then it is known that the 2-coordinate descent method satisfies [She and Schmidt, 2017]

$$\mathbb{E}[f(x^k)] - f(x^*) \leq \left(1 - \frac{\mu_2}{n^2 L_2}\right)^k (f(x^0) - f^*). \tag{12}$$

A similar result for a more-general problem class was shown by Necoara and Patrascu [2014]. This is substantially slower than the rate we show for the greedy 2-coordinate descent method. This rate is slower even in the extreme case where $\mu_1$ is similar to $\mu_2/n$, due to the presence of the $n^2$ term.

There also exist analyses for cyclic selection in the equality-constrained case but existing rates for cyclic rules are slower than the random rates Wang and Lin [2014].

In the case where $f$ is a dense quadratic function of $n$ variables, which includes SVMs under the most popular kernels, both random selection and greedy selection cost $O(n)$ per iteration to implement. If we consider the time required to reach an accuracy of $\epsilon$ under random selection using the rate (12) we obtain $O(n^3 \kappa \log(1/\epsilon))$ where $\kappa = L_2/\mu_2$. While for greedy selection under (11) it is between $O(n^2 \kappa \log(1/\epsilon))$ if $\mu_1$ is close to $\mu_2/n$ and $O(n \kappa \log(1/\epsilon))$ if $\mu_1$ is close to $\mu_2$. Thus, the reduction in total time complexity from using the greedy method is between a factor of $O(n)$ and $O(n^2)$. This is a large difference which has not been reflected in previous analyses.

There exist faster rates than (12) in the literature, but these require additional assumptions such as $f$ being separable or that we know the coordinate-wise Lipschitz constants [Necoara et al., 2011, Necoara and Patrascu, 2014, Necoara et al., 2017, Fang et al., 2018]. However, these assumptions restrict the applicability of the results. Further, unlike convergence rates for random coordinate selection, we note that the new linear convergence rate (11) for greedy 2-coordinate method avoids requiring a direct dependence on the problem dimension. The only previous dimension-independent convergence rate for the greedy 2-coordinate method that we are aware of is due to Beck [2014, Theorem 5.2b]. Their work considers functions that are bounded below, which is a weaker assumption than the proximal-PL assumption. However, this only leads to sublinear convergence rates and only on a measure of the violation in the Karush-Kuhn-Tucker conditions. Beck [2014, Theorem 6.2] also gives convergence rates in terms of function values for the special case of convex functions, but these rates are sublinear and dimension dependent.

## 3 Equality- and Bound-Constrained Greedy Coordinate Updates

Equality constraints often appear alongside lower and/or upper bounds on the values of the individual variables. This results in problems of the form

$$\min_{x \in \mathbb{R}^n} f(x), \quad \text{subject to } \sum_{i=1}^{n} x_i = \gamma, \ l_i \le x_i \le u_i. \tag{13}$$

This framework includes our motivating problems of optimizing over the probability simplex ($l_i = 0$ for all $i$ since probabilites are non-negative), and optimizing SVMs with an unregularized bias (where we have lower and upper bounds). With bound constraints we use a $d^k$ of form (3) but where $\delta^k$ is defined so that the step respects the constraints,

$$\delta^k = -\min \left\{ \frac{\alpha^k}{2} (\nabla_{i_k} f(x^k) - \nabla_{j_k} f(x^k)), x_{i_k}^k - l_{i_k}, u_{j_k} - x_{j_k}^k \right\}, \tag{14}$$

Unfortunately, analyzing the bound-constrained case is more complicated. There are several possible generalizations of the greedy rule for choosing the coordinates $i_k$ and $j_k$ to update, depending on what properties of (5) we want to preserve [see Nutini, 2018, Section 2.7]. In this section we discuss several possibilities, and how the choice of greedy rule affects the convergence rate and iteration cost.

### 3.1 GS-s: Minimizing Directional Derivative

Up until version 2.7, the greedy rule used in LIBSVM was the Gauss-Southwell-s (GS-s) rule. The GS-s rule chooses the coordinates resulting in the $d^k$ with the most-negative directional derivative. This is a natural generalization of the idea of steepest descent, and the first uses of the method that we aware of are by Keerthi et al. [2001] for SVMs and by Shevade and Keerthi [2003] for 1-norm regularized optimization. For problem (13) the GS-s rule chooses

$$i_k \in \underset{i \, | \, x_i^k > l_i}{\arg\max} \nabla_i f(x^k), \quad j_k \in \underset{j \, | \, x_j^k < u_i}{\arg\min} \nabla_j f(x^k). \tag{15}$$

This is similar to the unbounded greedy rule (5) but excludes variables where the update would immediately violate a bound constraint.

Unfortunately, the per-iteration decrease in $f$ obtained by the GS-s rule can be arbitrarily small. In particular, consider the case where the variable $i$ maximizing $\nabla_i f(x^k)$ has a value of $x_i^k = l_i + \epsilon$ for an arbitrarily small $\epsilon$. In this case, we would choose $i_k$ and take an arbitrarily small step of

$\delta^k = \epsilon$. Steps like this that truncate $\delta^k$ are called "bad" steps, and the GS-s rule does not guarantee a non-trivial decrease in $f$ on bad steps. If we only have bound constraints and do not have an equality constraint (so we can update on variable at a time), Karimireddy et al. [2019] show that at most half of the steps are bad steps. Their argument is that after we have taken a bad step on coordinate $i$, then the next time $i$ is chosen we will not take a bad step. However, with an equality constraint it is possible for a coordinate to be involved in consecutive bad steps. It is possible that a combinatorial argument similar to Lacoste-Julien and Jaggi [2015, Theorem 8] could bound the number of bad steps, but it is not obvious that we do not require an exponential total number of bad steps.

### 3.2 GS-q: Minimum 2-Coordinate Approximation

A variant of the Gauss-Southwell-q (GS-q) rule of Tseng and Yun [2009] for problem (13) is

$$\underset{i,j}{\arg\min} \ \underset{d_{ij}||d_i+d_j=0}{\min} \left\{ f(x^k) + \nabla_{ij} f(x^k)^T d_{ij} + \frac{1}{2\alpha^k} \|d_{ij}\|^2 : x^k + d \in [l, u] \right\}. \tag{16}$$

This minimizes a quadratic approximation to the function, restricted to the feasible set. For problem (13), the GS-q rule is equivalent to choosing $i_k$ and $j_k$ to maximize (14), the distance that we move. We show the following result for the GS-q rule in Appendix D.

**Theorem 3.1.** *Let $f$ be a differentiable function whose gradient is 2-coordinate-wise Lipschitz (7) and restricted to the set where $x^T 1 = \gamma$ and $l_i \leq x_i \leq u_i$. If this function satisfies the proximal-PL inequality in the 2-norm [Karimi et al., 2016] for some positive $\mu_2$, then the iterations of the 2-coordinate update (3) with $\delta^k$ given by (14), $\alpha^k = 1/L_2$, and the greedy GS-q rule (16) satisfy:*

$$f(x^k) - f(x^*) \leq \left( 1 - \frac{\mu_2}{L_2(n-1)} \right)^k (f(x^0) - f^*). \tag{17}$$

The proof of this result is more complicated than our previous results, relying on the concept of conformal realizations used by Tseng and Yun [2009]. We prove the result for general block sizes and then specialize to the two-coordinate case. Unlike the GS-s rule, this result shows that the GS-q guarantees non-trivial progress on each iteration. Note that while this result does have a dependence on the dimension $n$, it does not depend on $n^2$ as the random rate (12) does. Moreover, the dependence on $n$ can be improved by increasing the block size.

Unfortunately, the GS-q rule is not always efficient to use. As discussed by Beck [2014], there is no known algorithm faster than $O(n^2)$ for computing the GS-q rule (16). One special case where this can be solved in $O(n)$ given the gradient is if we only have lower bounds (or only have upper bounds) [Beck, 2014]. An example with only lower bounds is our motivating problem of optimizing over the probability simplex, which only requires variables to be non-negative and sum to 1. On the other hand, our other motivating problem of SVMs requires lower and upper bounds so computing the GS-q rule would require $O(n^2)$. Beginning with version 2.8, LIBSVM began using an approximation to the GS-q rule that can be computed in $O(n)$. In particular, LIBSVM first chooses one coordinate using the GS-s rule, and then optimizes the other coordinate according to a variant of the GS-q rule [Fan et al., 2005].[1] While other rules have been proposed, the LIBSVM rule remains among the best-performing methods in practice [Horn et al., 2018]. However, similar to the GS-s rule we cannot guarantee non-trivial progress for the practical variant of the GS-q rule used by LIBSVM.

### 3.3 GS-1: Steepest Descent in the 1-Norm

Rather than using the classic GS-s or GS-q selection rules, the Gauss-Southwell-1 (GS-1) rule performs steepest descent in the 1-norm. For problem (13) this gives the update

$$d^k \in \underset{l_i \leq x_i + d_i \leq u_i | d^T 1 = 0}{\arg\min} \left\{ \nabla f(x^k)^T d + \frac{1}{2\alpha^k} \|d\|_1^2 \right\}. \tag{18}$$

The GS-1 rule was proposed by Song et al. [2017] for (unconstrained) 1-norm regularized problems. To analyze this method, we modify the definition of $\mathcal{D}(x, L)$ in the proximal-PL assumption to be

$$\mathcal{D}(x, L) = -2L \underset{\{l_i \leq y_i \leq u_i \ | y^T 1 = \gamma\}}{\min} \left\{ \langle \nabla f(x), y - x \rangle + \frac{L}{2} \|y - x\|_1^2 \right\}. \tag{19}$$

We then have the following dimension-independent convergence rate for the GS-1 rule.

---

[1] The newer LIBSVM rule also uses Lipschitz information about each coordinate; see Section 4 for discussion.

**Algorithm 1** The GS-1 algorithm (with variables sorted in descending order according to $\nabla f(x)$).

```
1: function GS-1(x, ∇f(x), α, l, u)
2:     x_0 ← 0; x_{n+1} ← 0; i ← 1; j ← n; d ← 0;
3:     while 1 do
4:         δ ← α/4 (∇_i f(x) − ∇_j f(x))
5:         ω = Σ_{p=0}^{i-1} x_p − l_p; κ = Σ_{q=j+1}^{n+1} u − x_q
6:         if δ − ω < 0  &  δ − κ < 0 then
7:             if ω < κ then d_i = ω − κ ; break;
8:             else d_j = ω − κ; break;
9:             end if
10:        else if δ − ω < 0 then d_j = ω − κ; break;
11:        else if δ − κ < 0 then d_i = ω − κ; break;
12:        end if
13:        if x_i + ω − δ ≥ l_i  &  x_j − κ + δ ≤ u_j then
14:            d_i = ω − δ; d_j = δ − κ; break;
15:        end if
16:        if x_i + ω − δ < l_i  &  x_j − κ + δ > u_j then
17:            if l_i − (x_i + ω − δ) > x_j − κ + δ − u_j then
18:                d_i = l − x_i; i ← i + 1
19:            else
20:                d_j = u − x_j; j ← j − 1
21:            end if
22:        else if x_i + ω − δ < l_i then d_i = l − x_i; i ← i + 1
23:        else d_j = u − x_j; j ← j − 1
24:        end if
25:    end while
26:    return d
27: end function
```

**Theorem 3.2.** *Let $f$ be a differentiable function whose gradient is 2-coordinate-wise Lipschitz* (7) *and restricted to the set where $x^T 1 = \gamma$ and $l_i \leq x_i \leq u_i$. If this function satisfies the proximal-PL inequality in the 1-norm* (9) *for some positive $\mu_1$ with the definition* (19)*, then the iterations of the update $x^{k+1} = x^k + d^k$ with the greedy rule* (18) *and $\alpha_k = 1/L_1 = 2/L_2$ satisfy:*

$$f(x^k) - f(x^*) \leq \left(1 - \frac{2\mu_1}{L_2}\right)^k (f(x^0) - f^*). \tag{20}$$

*Proof.* The proof follows the same reasoning as Theorem 2.3, but beginning after the application of Lemma 2.1 since we are directly computing the steepest descent direction. $\square$

This GS-1 convergence rate is at least as fast as the convergence rate for GS-q, and thus by exploiting a connection to the 1-norm we once again obtain a faster dimension-independent rate. In Algorithm 1 we give a method to construct a solution to the GS-1 rule (18) in $O(n \log n)$ time (due to sorting the $\nabla_i f(x^k)$ values). Thus, our new GS-1 update guarantees non-trivial progress at each step (unlike the GS-s rule) and is efficient to compute (unlike the GS-q rule). The precise logic of Algorithm 1 is somewhat complicated, but it can intuitively be viewed as a version of GS-s that fixes the bad steps where $\delta^k$ is truncated. Roughly, if the GS-s rule gives a bad step then the GS-1 moves the violating variable to its boundary and then may also update the variable with the next largest/smallest $\nabla_i f(x^k)$.

The drawback of the GS-1 update is that it is not strictly a 2-coordinate method. While the GS-1 update moves at most 2 variables within the interior of the bound constraints, it may move additional variables to their boundary. The iteration cost of the method will be higher on iterations where more than 2 variables are updated. However, by using an argument similar to Sun et al. [2019], we can show that the GS-1 rule will only update more than 2 variables on a finite number of early iterations. This is because, after some finite number of iterations, the variables actively constrained by their bounds will remain at their bounds. At this point, each GS-1 update will only update 2 variables within the

interior of the bounds. In the case of SVMs, moving a variable to its lower bound corresponds to removing it as a potential support vector. Thus, this "bug" of GS-1 that it may update more than 2 variables can allow it to quickly remove many support vectors. In our experiments, we found that GS-1 identified the support vectors more quickly than other rules and that most GS-1 updates only updated 2 or 3 coordinates.

## 4 Greedy Updates using Coordinate-Wise Lipschitz Constants

Up until this point, we have measured smoothness based on the maximum blockwise Lipschitz-constant $L_2$. An alternative measure of smoothness is Lipschitz continuity of individual coordinates. In particular, coordinate-wise Lipschitzness of coordinate $i$ requires that for all $x$ and $\alpha$

$$|\nabla_i f(x + \alpha e_i) - \nabla_i f(x)| \leq L_i |\alpha|,$$

where $e_i$ is a vector with a one in position $i$ and zeros in all other positions. For twice-differentiable convex functions, the Lipschitz constant with respect to the block $(i, j)$ is upper bounded by the sum of the coordinate-wise constants $L_i$ and $L_j$ [Nesterov, 2012, Lemma 1]. For equality-constrained optimization, Necoara et al. [2011] uses the coordinate-wise Lipschitz constants to design sampling distributions for $i_k$ and $j_k$. Their analysis gives rates that can be faster than uniform sampling (12).

In Appendix E, we consider greedy rules that depend on the $L_i$ values for the equality-constrained case. In particular, we show that the equality-constrained GS-q rule chooses $i_k$ and $j_k$ by solving

$$\arg\max_{i,j} \left\{ \frac{(\nabla_i f(x) - \nabla_j f(x))^2}{L_i + L_j} \right\}, \tag{21}$$

which yields the standard greedy rule (5) if all $L_i$ values are equal. We show that the coordinate descent update with this selection rule and

$$\delta^k = -(\nabla_{i_k} f(x^k) - \nabla_{j_k} f(x^k))/(L_{i_k} + L_{j_k}), \tag{22}$$

can be written as steepest descent in the norm defined by $||d||_L \triangleq \sum_i \sqrt{L_i} |d_i|$. This yields a convergence rate that can be faster than the greedy rate (11).

Unfortunately, it is not obvious how to solve (21) faster than $O(n^2)$. Nevertheless a reasonable approximation is to use

$$i_k \in \arg\max_i \nabla_i f(x^k)/\sqrt{L_i}, \quad j_k \in \arg\min_j \nabla_j f(x^k)/\sqrt{L_j}. \tag{23}$$

which we call the ratio approximation. This approximation is (21) after re-parameterizing in terms of variables $x_i/\sqrt{L_i}$ so that all coordinate-wise Lipschitz constants are 1 in the transformed problem. We can also use this re-parameterization to implement variations of the GS-s/GS-q/GS-1 rules if we also have bound constraints. While the ratio approximation (23) performed nearly as well as the more expensive (21) in our experiments, we found that the gap could be improved slightly if we choose one coordinate according to the ratio approximation and then the second coordinate to optimize (21).[2]

## 5 Experiments

Our first experiment evaluates the performance of various rules on a synthetic equality-constrained least squares problem. Specifically, the objective is $f(x) = \frac{1}{2}||Ax - b||^2$ subject to $x^T 1 = 0$. We generate the elements of $A \in \mathbb{R}^{1000 \times 1000}$ from a standard normal and set $b = Ax + z$ where $x$ and $z$ are generated from standard normal distributions. We also consider a variant where each column of $A$ is scaled by a sample from a standard normal to induce very-different $L_i$ values. In Figure 1 we compare several selection rules: random $i_k$ and $j_k$, the greedy rule (5), sampling $i_k$ and $j_k$ proportional to $L_i$, the exact greedy $L_i$ rule (21), the ratio greedy $L_i$ rule (23), and a variant where we set one coordinate using (23) and other using (21) (switching between the two). All algorithms use the update (22). In these experiments we see that greedy rules lead to faster convergence than random rules in all cases. We see that knowing the $L_i$ values does not significantly change the performance of the random method, nor does it change the performance of the greedy methods in the case when

---

[2]This strategy is similar to LIBSVM's rule beginning in version 2.8 for the special case of quadratic functions.

the $L_i$ were similar. However, with different $L_i$ the (expensive) exact greedy method exploiting $L_i$
works much better. We found that the ratio method worked similar to or better than the basic greedy
method (depending on the random seed), while the switching method often performed closer to the
exact method.

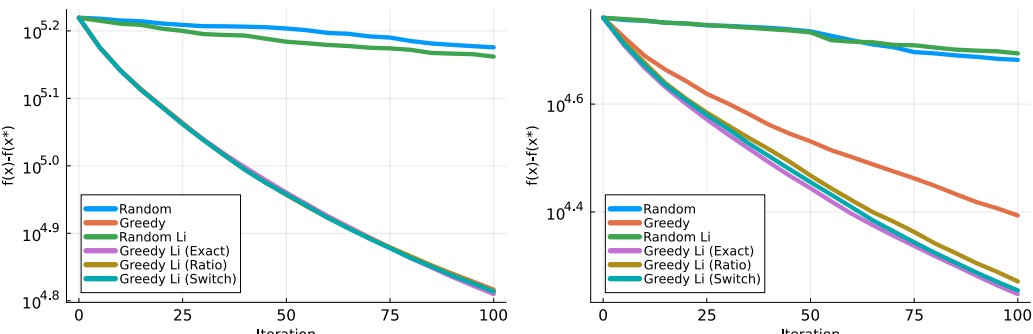

Figure 1: Random vs greedy coordinate selection rules, including rules using the coordinate-wise
Lipschitz constants $L_i$. The $L_i$ are similar in the left plot, but differ significantly on the right.

Our second experiment considers the same problem but with the additional constraints $x_i \in [-1, 1]$.
Figure 2 compares the GS-s, GS-q, and GS-1 rules in this setting. We see that the GS-s rule results in
the slowest convergence rate, while the GS-q rule rule takes the longest to identify the active set. The
GS-1 rule typically updates 2 or 3 variables, but on early iterations it updates up to 5 variables.

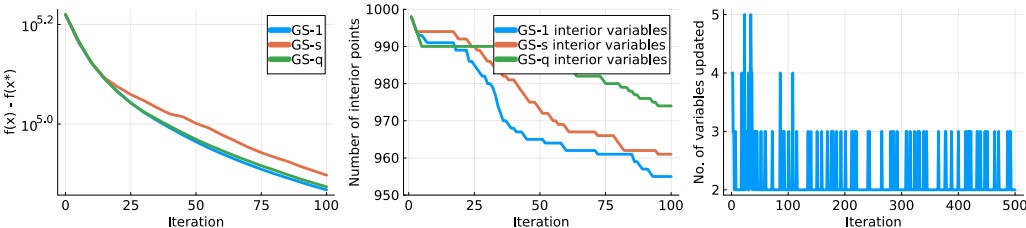

Figure 2: Comparison of GS-1, GS-q and GS-s under linear equality constraint and bound constraints.
The left plot shows the function values, the middle plot shows the number of interior variables, and
the right plot shows the number of variables updated by the GS-1 rule.

# 6 Discussion

Despite the popularity of LIBSVM, up until this work we did not have a strong justification for using
greedy 2-coordinate methods over simpler random 2-coordinate methods for equality-constrained
optimization methods. This work shows that greedy methods may be faster by a factor ranging from
$O(n)$ up to $O(n^2)$. This work is the first to identify the equivalence between the greedy 2-coordinate
update and steepest descent in the 1-norm. The connection to the 1-norm is key to our simple analyses
and also allows us to analyze greedy rules depending on coordinate-wise Lipschitz constants.

For problems with bound constraints and equality constraints, we analyzed the classic GS-q rule but
also proposed the new GS-1 rule. Unlike the GS-s rule the GS-1 rule guarantees non-trivial progress
on each iteration, and unlike the GS-q rule the GS-1 rule can be implemented in $O(n \log n)$. We
further expect that the GS-1 rule could be implemented in $O(n)$ by using randomized algorithms,
similar to the techniques used to implement $O(n)$-time projection onto the 1-norm ball Duchi et al.
[2008], van den Berg et al. [2008]. The disadvantage of the GS-1 rule is that on some iterations it may
update more than 2 coordinates on each step. However, when this happens the additional coordinates
are simply moved to their bound. This can allow us to identify the active set of constraints more
quickly. For SVMs this means identifying the support vectors faster, giving cheaper iterations.

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
