# A Equivalent Ways of Writing Equality-Constrained Greedy Rule

365 We first show that the greedy rule with a summation constraint, of choosing the max/min partial
366 derivatives, is an instance of the GS-q rule. We then show that this rule is also equivalent to steepest
367 descent in the 1-norm.

## A.1 Greedy Rule Maximizes GS-q Progress Bound

369 For the optimization problem (1), the GS-q rule selects the optimal block $b = \{i, j\}$, by solving the
370 following minimization problem:

$$b = \arg\min_b \left\{ \min_{d_b | d_i + d_j = 0} \langle \nabla_b f(x), d_b \rangle + \frac{1}{2\alpha} ||d_b||^2 \right\}, \tag{24}$$

371 where $d_b$ is the descent direction.

**Solving for $d_b$.** First let us fix $b$ and solve for $d_b$. The Lagrangian of (24) is,

$$\mathcal{L}(d_b, \lambda) = \langle \nabla_b f(x), d_b \rangle + \frac{1}{2\alpha} ||d_b||^2 + \lambda(d_1 + d_2).$$

373 Taking the gradient with respect to $d_b$ gives,

$$\nabla_{d_b} \mathcal{L}(d_b, \lambda) = \nabla_b f(x) + \frac{1}{\alpha} d_b + \lambda 1.$$

374 Setting the gradient equal to 0 and solving for $d_b$ gives,

$$d_b = -\alpha(\nabla_b f(x) + \lambda 1). \tag{25}$$

375 From our constraint, $d_i + d_j = 0$, we get

$$0 = -\alpha \left( \nabla_i f(x) + \lambda + \nabla_j f(x) + \lambda \right),$$

$$\lambda = -\frac{1}{2} \langle \nabla_b f(x), 1 \rangle.$$

376 Substituting in (25) we get,

$$d_b = -\alpha \left( \nabla_b f(x) - \frac{1}{2} \langle \nabla_b f(x), 1 \rangle 1 \right). \tag{26}$$

377 This can be re-written as

$$\begin{bmatrix} d_i \\ d_j \end{bmatrix} = \frac{\alpha}{2} \left( \nabla_i f(x) - \nabla_j f(x) \right) \begin{bmatrix} -1 \\ 1 \end{bmatrix}.$$

**Solving for $b$.** Now, we plug in the optimal $d_b$ from (26) in (24) and solve for $b$ to give

$$\arg\min_b -\alpha \left\langle \nabla_b f(x), (\nabla_b f(x) - \frac{1}{2} \langle \nabla_b f(x), 1 \rangle 1) \right\rangle + \frac{\alpha}{2} ||(\nabla_b f(x) - \frac{1}{2} \langle \nabla_b f(x), 1 \rangle 1)||^2$$

$$\equiv \arg\min_b -||\nabla_b f(x)||^2 + \frac{1}{2} (\langle \nabla_b f(x), 1 \rangle)^2 + \frac{1}{2} ||\nabla_b f(x)||^2 - \frac{1}{2} (\langle \nabla_b f(x), 1 \rangle)^2 +$$

$$\frac{1}{8} (\langle \nabla_b f(x), 1 \rangle)^2 \underbrace{\langle 1, 1 \rangle}_{2}$$

$$\equiv \arg\min_b -\frac{1}{2} ||\nabla_b f(x)||^2 + \frac{1}{4} (\langle \nabla_b f(x), 1 \rangle)^2$$

$$\equiv \arg\max_b ||\nabla_b f(x)||^2 - \frac{1}{2} (\langle \nabla_b f(x), 1 \rangle)^2$$

$$\equiv \arg\max_b ||\nabla_b f(x)||^2 - \frac{1}{2} (\nabla_i f(x) + \nabla_j f(x))^2$$

$$\equiv \arg\max_b \frac{1}{2} ||\nabla_b f(x)||^2 - \nabla_i f(x) \nabla_j f(x)$$

$$\equiv \arg\max_b \frac{1}{2} (\nabla_i f(x) - \nabla_j f(x))^2)$$

$$\equiv \arg\max_b |\nabla_i f(x) - \nabla_j f(x)|. \tag{27}$$

Therefore, the GS-q rule chooses the $i$ and $j$ that are farthest apart, which are the coordinates with maximum and minimum values in $\nabla f(x)$.

## A.2 Greedy Rule is Steepest Descent in the 1-Norm (Lemma 2.1)

The steepest descent method finds the descent direction that minimizes the function value in every iteration. That is,

$$d = \underset{d \in \mathbb{R}^n \mid d^T 1 = 0}{\arg\min} \left\{ \nabla f(x)^T d + \frac{1}{2\alpha} \|d\|_1^2 \right\}. \tag{28}$$

The proof follows by constructing a solution to the steepest descent problem (28) which only has two non-zero entries. The Lagrangian of (28) is,

$$\mathcal{L}(d, \lambda) = \nabla f(x)^T d + \frac{1}{2\alpha} \|d\|_1^2 + \lambda d^T 1.$$

The sub-differential with respect to $d$ and $\lambda$ is given by

$$\partial_d \mathcal{L}(d, \lambda) \equiv \nabla f(x) + \frac{1}{2\alpha} \partial \|d\|_1^2 + \lambda 1,$$

$$\partial_\lambda \mathcal{L}(d, \lambda) \equiv d^T 1.$$

We have that the zero vector is an element of the sub-differential at the solution. From $0 \in \partial_\lambda \mathcal{L}(d, \lambda)$ we have $d^T 1 = 0$. From $0 \in \partial_d \mathcal{L}(d, \lambda)$ at the solution we require

$$2\alpha(-\nabla f(x) - \lambda 1) \in \partial \|d\|_1^2,$$

or equivalently by using that $\partial_i \|d\|_1^2 \equiv 2\|d\|_1 \mathrm{sgn}(d_i)$ this subgradient inclusion is equivalent to having for each coordinate $i$ that

$$\alpha(-\nabla_i f(x) - \lambda 1) = \|d\|_1 \mathrm{sgn}(d_i), \tag{29}$$

where the signum function $\mathrm{sgn}(d_i)$ is $+1$ if $d_i$ is positive, $-1$ if $d_i$ is negative, and can take any value in the interval $[-1, 1]$ if $d_i$ is zero.

Let $i \in \arg\max_i \{\nabla_i f(x)\}$ and $j \in \arg\min_j \{\nabla_j f(x)\}$. Consider a solution $d$ such that $d_i = \delta, d_j = -\delta$ for some $\delta \in \mathbb{R}$ and $d_k = 0$ for if $k \neq i$ and $k \neq j$. By construction the vector $d$ has only two non-zero coordinates and satisfies the sum-to-zero constraint required for feasibility. Thus, we have a solution if we can choose $\delta$ to satisfy (29) for all coordinates.

The definition of $d$ implies $\|d\|_1 = 2\delta$, while $\mathrm{sgn}(d_i) = 1$, $\mathrm{sgn}(d_j) = -1$ and $\mathrm{sgn}(d_k) \in [-1, 1]$. Thus, for $d$ to be a steepest descent direction we must have:

$$-\alpha \nabla_i f(x) - \alpha \lambda = 2\delta \tag{30}$$
$$-\alpha \nabla_j f(x) - \alpha \lambda = -2\delta \tag{31}$$
$$-\alpha \nabla_k f(x) - \alpha \lambda \in 2\delta[-1, 1]. \tag{32}$$

Solving for $\lambda$ in (30) gives

$$\lambda = -\nabla_i f(x) - 2\delta/\alpha, \tag{33}$$

and substituting this in (31) gives,

$$\delta = -\frac{\alpha}{4}(\nabla_i f(x) - \nabla_j f(x)). \tag{34}$$

It remains only to show that (32) is satisfied by $d$. Using the value of $\lambda$ (33) in (32) yields,

$$-\alpha \nabla_k f(x) + \alpha \nabla_i f(x) + 2\delta \in 2\delta[-1, 1].$$

Now, substituting the value for $\delta$ (34) gives

$$-\alpha \nabla_k f(x) + \alpha \nabla_i f(x) - \frac{\alpha}{2}(\nabla_i f(x) - \nabla_j f(x)) \in -\frac{\alpha}{2}(\nabla_i f(x) - \nabla_j f(x))[-1, 1],$$

and multiplying by $2/\alpha$ this is equivalent to

$$-2\nabla_k f(x) + \nabla_i f(x) + \nabla_j f(x) \in -(\nabla_i f(x) - \nabla_j f(x))[-1, 1],$$

which can be satisfied for some value in $[-1, 1]$ if

$$-2\nabla_k f(x) + \nabla_i f(x) + \nabla_j f(x) \le |\nabla_i f(x) - \nabla_j f(x)|.$$

As $\nabla_k f(x)$ is between $\nabla_i f(x)$ and $\nabla_j f(x)$, we can write it as a convex combination $\theta \nabla_i f(x) + (1-\theta)\nabla_j f(x)$ for some $\theta \in [0,1]$. Thus, we require

$$-2(\theta \nabla_i f(x) + (1-\theta)\nabla_j f(x)) + \nabla_i f(x) + \nabla_j f(x)$$
$$= (1 - 2\theta)(\nabla_i f(x) - \nabla_j f(x)) \le |\nabla_i f(x) - \nabla_j f(x)|,$$

which holds because $(1 - 2\theta) \in [-1, 1]$.

We have shown that a two-coordinate update $d$ satisfies the sufficient conditions to be a steepest descent direction in the 1-norm. Substituting $d$ back into the expression for steepest descent gives

$$\min_{d \in \mathbb{R}^n | d^T 1 = 0} \nabla f(x)^T d + \frac{1}{2\alpha}||d||_1^2 = \nabla_{ij} f(x)^T d_{ij} + \frac{1}{2\alpha}||d_{ij}||_1^2$$

$$\ge \min_{i,j} \left\{ \min_{d_{i,j} \in \mathbb{R}^2 | d_i + d_j = 0} \nabla_{ij} f(x)^T d_{ij} + \frac{1}{2\alpha}||d_{ij}||_1^2 \right\}.$$

The reverse inequality follows from the fact that a two coordinate update cannot lead to a smaller value than updating all coordinates, so we have

$$\min_{d \in \mathbb{R}^n | d^T 1 = 0} \nabla f(x)^T d + \frac{1}{2\alpha}||d||_1^2 = \min_{i,j} \left\{ \min_{d_{i,j} \in \mathbb{R}^2 | d_i + d_j = 0} \nabla_{ij} f(x)^T d_{ij} + \frac{1}{2\alpha}||d_{ij}||_1^2 \right\}.$$

## B   Relating Lipschitz Constants

**Proposition B.1.** *Suppose $f$ is twice differentiable and*

$$\sup_{x:\langle x,1 \rangle = a} \max_d \left\{ d^\top \nabla^2 f(x) d : \langle d, 1 \rangle = 0, supp(d) = 2, ||d||_1 \le 1 \right\} = L_1. \tag{35}$$

*Then $f$ satisfies the following inequality:*

$$f(x + d) \le f(x) + \langle \nabla f(x), d \rangle + \frac{L_1}{2}||d||_1^2, \tag{36}$$

*for $x$ such that $\langle x, 1 \rangle = a$ and any $d$ such that $\langle d, 1 \rangle = 0$. That is, $f$ is full-coordinate Lipschitz smooth in the $\ell_1$ norm with constant $L_1$.*

*Proof.* Consider the optimization problem

$$\max_d \left\{ d^\top \nabla^2 f(x) d : \langle d, 1 \rangle = 0, ||d||_1 \le 1 \right\}. \tag{37}$$

We will show that the maximum is achieved by at least one $d$ satisfying $d_i = -d_j \ne 0$, $d_k = 0$ for all $k \ne i, j$. That is, a two coordinate update achieves the maximum.

First, observe that Equation (37) is a convex maximization problem over a (convex) polyhedron. As a result, at least one solution occurs at an extreme point of the constraint set,

$$\mathcal{D} = \{d : \langle d, 1 \rangle = 0, ||d||_1 \le 1\}.$$

The proof proceeds by showing that all extreme points of $\mathcal{D}$ contain exactly two non-zero entries. Let $d_e$ be any extreme point of $\mathcal{D}$ and suppose by way of contradiction that $d_e$ has at least three non-zero entries. Denote these entries as $d_1, d_2, d_3$. Since at least one entry of $d_e$ must be negative and one must be positive, we may assume without loss of generality that $d_1, d_2 > 0$ and $d_3 < 0$.

Let $\epsilon > 0$ and define $d'_e = d_e + e_1 \epsilon - e_2 \epsilon$. For $\epsilon$ sufficiently small it holds that $d_1 + \epsilon > 0$ and $d_2 - \epsilon > 0$ so that

$$(d_1 + \epsilon) + (d_2 - \epsilon) + d_3 = d_1 + d_2 + d_3.$$

428 We conclude

$$\begin{aligned}
\|d'_e\|_1 &= |d_1 + \epsilon| + |d_2 - \epsilon| + |d_3| \\
&= (d_1 + \epsilon) + (d_2 - \epsilon) + |d_3| \\
&= |d_1| + |d_2| + |d_3| \\
&= d_1 + d_2 + d_3 \\
&= \|d_e\|_1 \\
&\le 1.
\end{aligned}$$

429 Thus, $d'_e \in \mathcal{D}$. Define $d''_e = d_e - e_1\epsilon + e_2\epsilon$ and observe $d''_e \in \mathcal{D}$ by a symmetric argument. Moreover,

$$d_e = \frac{1}{2}d'_e + \frac{1}{2}d''_e,$$

430 i.e. the extreme point is a convex combination of two points in $\mathcal{D}$. This contradicts the definition of
431 an extreme point, so we have proved that every extreme point of $\mathcal{D}$ has at most two non-zero entries
432 Since no point of $\mathcal{D}$ can have exactly one non-zero entry and $0$ is the relative interior of $\mathcal{D}$, we have
433 shown every extreme point has exactly two non-zero entries.

434 As a result, (37) is maximized at at least one extreme point $d_e$, where $\mathrm{supp}(d_e) = 2$. Thus, we may
435 restrict optimization to directions of support two, giving

$$\begin{aligned}
\max_d \ &\left\{ d^\top \nabla^2 f(x) d : \langle d, 1 \rangle = 0, \|d\|_1 \le 1 \right\} \\
&= \max_d \left\{ d^\top \nabla^2 f(x) d : \langle d, 1 \rangle = 0, \mathrm{supp}(d) = 2, \|d\|_1 \le 1 \right\} \\
&\le L_1.
\end{aligned}$$

436 It is now straightforward to obtain the final result using a Taylor expansion and the Lagrange form of
437 the remainder. In particular, for some parameter $x'\mathrm{Conv}(\{x, x+d\})$ we have

$$\begin{aligned}
f(x+d) &= f(x) + \langle \nabla f(x), d \rangle + \frac{1}{2}d^\top \nabla^2 f(x + \alpha d)d \\
&\le f(x) + \langle \nabla f(x), d \rangle + \frac{1}{2}\|d\|_1^2 \max_v \left\{ v^\top \nabla^2 f(x')v : \langle v, 1 \rangle = 0, \|v\|_1 \le 1 \right\} \\
&= f(x) + \langle \nabla f(x), d \rangle + \frac{1}{2}\|d\|_1^2 \max_v \left\{ v^\top \nabla^2 f(x')v : \langle v, 1 \rangle = 0, \mathrm{supp}(v) = 2, \|v\|_1 \le 1 \right\} \\
&= f(x) + \langle \nabla f(x), d \rangle + \frac{L_1}{2}\|d\|_1^2,
\end{aligned}$$

438 which gives the result. $\qquad\square$

439 **Proposition B.2.** *The constant $L_1$ in (35) is exactly equal to $\frac{L_2}{2}$.*

440 *Proof.* Let $d \in \mathbb{R}^n$ such that $\mathrm{supp}(d) = 2$ and $\langle d, 1 \rangle = 0$. WLOG, suppose that the two non-zero
441 entries of $d$ are $d_1$ and $d_2$. Observe that $\langle d, 1 \rangle = 0$ implies $d_1 = -d_2$ and $\|d\|_1 = \sqrt{2}\|d\|_2$. Thus we
442 have

$$\begin{aligned}
L_2 &= \sup_{x:\langle x,1\rangle = a} \max_d \left\{ d^\top \nabla^2 f(x)d : \langle d, 1 \rangle = 0, \mathrm{supp}(d) = 2, \|d\|_2 \le 1 \right\} \\
&= 2 \sup_{x:\langle x,1\rangle = a} \max_d \left\{ d^\top \nabla^2 f(x)d : \langle d, 1 \rangle = 0, \mathrm{supp}(d) = 2, \|d\|_1 \le 1 \right\} \\
&= 2L_1,
\end{aligned}$$

443 where we have used Proposition B.3 to relate the variational characterizations to the Lipschitz
444 constants in question. This completes the proof.

445 $\qquad\square$

446 **Proposition B.3.** *Let $\|\cdot\|$ an arbitrary norm and define the dual norm on the feasible space,*

$$\|v\|_* = \sup \left\{ z^\top v : \langle z, 1 \rangle = 0, supp(z) = 2, \|z\| \le 1 \right\}.$$

447   *Then the variational characterization based on the Hessian,*

$$L = \sup_{x:\langle x,1\rangle=a} \max_d \left\{ d^\top \nabla^2 f(x) d : \langle d,1\rangle = 0, supp(d) = 2, \|d\| \le 1 \right\},$$

448   *gives the two-coordinate Lipschitz constant of $\nabla f$ (see Equation (7)) in norm $\|\cdot\|$ on the feasible*
449   *space.*

450   *Proof.* Let $x$ be feasible (i.e. $\langle x,1\rangle = a$) and define

$$\mathcal{D} = \{d : \langle d,1\rangle = 0, \operatorname{supp}(d) = 2, \|d\|_1 \le 1\}.$$

451   Suppose $d$ is some be feasible 2-coordinate update, not necessarily unit norm. The fundamental
452   theorem of calculus implies

$$\nabla_{ij} f(x + d) - \nabla_{ij} f(x) = \int_0^1 \nabla_{ij}^2 f(x + td) d\, dt$$

453   Taking norms on both sides, we obtain

$$\begin{aligned}
\|\nabla_{ij} f(x + d) - \nabla_{ij} f(x)\|_* &= \| \int_0^1 \nabla_{ij}^2 f(x + td) d\, dt \|_* \\
&\le \int_0^1 \|\nabla_{ij}^2 f(x') d\|_* dt \\
&\le \|d\| \int_0^1 \sup_{d' \in \mathcal{D}} \left\{ d'^\top \nabla_{ij}^2 f(x + td) d' \right\} dt \\
&\le L\|d\|,
\end{aligned}$$

454   where we have used the definition of the dual norm. For the reverse inequality, let $\tilde{L}$ be the Lipschitz
455   constant of $\nabla f$ in norm $\|\cdot\|$. Observe that for any feasible $x$ and 2-coordinate update $d$, there exists
456   $\alpha \in (0,1)$ and $\tilde{x} = x + \alpha d$ such that

$$\nabla_{ij}^2 f(\tilde{x}) d = \nabla_{ij} f(x + d) - \nabla_{ij} f(x).$$

457   Using this, we obtain

$$\begin{aligned}
d^\top \nabla_{ij}^2 f(\tilde{x}) d &\le \|d\|\|\nabla_{ij}^2 f(\tilde{x}) d\|_* \\
&= \|d\|\|\nabla_{ij} f(x + d) - \nabla_{ij} f(x)\|_* \\
&\le \tilde{L}\|d\|^2.
\end{aligned}$$

458   Dividing by sides by $\|d\|^2$, taking $\|d\| \to 0$, and supremizing over $x, d$ gives

$$L = \sup_{x:\langle x,1\rangle=a} \max_{d \in \mathcal{D}} \left\{ d^\top \nabla^2 f(x) d \right\} \le \tilde{L}$$

459   We conclude $\tilde{L} = L$ as desired.     □

## C   Relationship Between Proximal-PL Constants

461   **Lemma C.1.** *Suppose that $F(x) = f(x) + g(x)$ satisfies the proximal-PL inequality in the $\ell_2$-norm*
462   *with constants $L_2, \mu_2$. Then $F$ also satisfies the proximal-PL inequality in the $\ell_1$-norm with constants*
463   *$L_1$ and $\mu_1 \in [\mu_2/n, \mu_2]$.*

464   *Proof.* Proximal-PL inequality in the $\ell_2$-norm implies

$$\begin{aligned}
F(x) - F(x^*) &\le -\frac{L_2}{\mu_2} \min_y \left\{ \langle \nabla f(x), y - x \rangle + \frac{L_2}{2}\|y - x\|_2^2 + g(y) - g(x) \right\} \\
&\le -\frac{L_2}{\mu_2} \min_y \left\{ \langle \nabla f(x), y - x \rangle + \frac{L_2}{2n}\|y - x\|_1^2 + g(y) - g(x) \right\} \\
&\le -\frac{L_2 L_1 n}{L_2 \mu_2} \min_y \left\{ \langle \nabla f(x), y - x \rangle + \frac{L_1}{2}\|y - x\|_1^2 + g(y) - g(x) \right\} \\
&= -\frac{L_1 n}{\mu_2} \min_y \left\{ \langle \nabla f(x), y - x \rangle + \frac{L_1}{2}\|y - x\|_1^2 + g(y) - g(x) \right\},
\end{aligned}$$

where the last inequality follows from Karimireddy et al. [2018][Lemma 9] with the choice of $\beta = \frac{L_2}{L_1 n}$, $h(y) = \langle \nabla f(x), y - x \rangle + g(y) - g(x)$, and $V(y) = \sqrt{L_2/2n}\|y - x\|_1$. Note that $\beta \in (0, 1]$ since $L_1 n \geq L_2$ and $h(x) = V(x) = 0$ so that the conditions of the lemma are satisfied. We conclude that proximal-PL inequality holds with $\mu_1 \geq \mu_2/n$.

We establish the reverse direction similarly; starting from proximal-PL in the $\ell_1$-norm,

$$
\begin{aligned}
F(x) - F(x^*) &\leq -\frac{L_1}{\mu_1} \min_y \left\{ \langle \nabla f(x), y - x \rangle + \frac{L_1}{2}\|y - x\|_1^2 + g(y) - g(x) \right\} \\
&\leq -\frac{L_1}{\mu_1} \min_y \left\{ \langle \nabla f(x), y - x \rangle + \frac{L_1}{2}\|y - x\|_2^2 + g(y) - g(x) \right\} \\
&\leq -\frac{L_1 L_2}{L_1 \mu_1} \min_y \left\{ \langle \nabla f(x), y - x \rangle + \frac{L_2}{2}\|y - x\|_2^2 + g(y) - g(x) \right\} \\
&= -\frac{L_2}{\mu_1} \min_y \left\{ \langle \nabla f(x), y - x \rangle + \frac{L_2}{2}\|y - x\|_2^2 + g(y) - g(x) \right\},
\end{aligned}
$$

where now we have used the same lemma with $V(y) = \sqrt{L_1/2}\|y - x\|_2$ and $\beta = \frac{L_1}{L_2}$, noting that $\beta \in (0, 1]$ since $L_1 \leq L_2$. This shows that $\mu_2 \geq \mu_1$, which completes the proof. $\qquad\square$

# D   Analysis of GS-q for Bound-Constrained Problem

In this section, we show linear convergence of greedy 2-coordinate descent under a linear equality constraint and bound constraints for the problem in (13) when using the GS-q rule. First, we introduce two definitions which underpin the theoretical machinery used in this section.

**Definition D.1** (Conformal Vectors). Let $d, d' \in \mathbb{R}^n$. We say that $d'$ is conformal to $d$ if

$$\{i : d_i' \neq 0\} \subseteq \{i : d_i \neq 0\},$$

that is, the support of $d'$ is a subset of the support of $d$, and $d_i d_i' \geq 0$ for every $i \in \{1, \ldots n\}$.

**Definition D.2** (Elementary Vector). Let $S \subset \mathbb{R}^n$ be a subspace. A vector $d \in \mathcal{S}$ is an elementary vector of $\mathcal{S}$ if there does not exist $d'$ conformal to $d$ with strictly smaller support, that is

$$\{i : d_i' \neq 0\} \subsetneq \{i : d_i \neq 0\}.$$

With these definitions in hand, we can state Lemma D.3, which is the key property we use in our proof strategy.

**Lemma D.3** (Conformal Realizations). *Let $S$ be a subspace of $\mathbb{R}^n$ and $t = \min_{x \in S} supp(x)$. Let $\tau \in \{t, \ldots, n\}$. Then every non-zero vector $x$ of $S \subseteq \mathbb{R}^n$ can be realized as the sum*

$$x = d_1 + \cdots + d_s + d_{s+1},$$

*where $d_1, \ldots, d_s$ are elementary vectors of $S$ that are conformal to $x$ and $d_{s+1} \in S$ is a vector conformal to $x$ with $supp(d_{s+1}) = \tau$. Furthermore, $s \leq n - \tau$.*

We include a proof in Appendix D.1; see Tseng and Yun [2009, Proposition 6.1] for an alternative (earlier) statement and proof. Using this tool, we prove the following convergence rate for 2-coordinate descent with the GS-q rule.

**Theorem D.4.** *Let the function $F(x) = f(x) + h(x)$, where $f : \mathbb{R}^n \to \mathbb{R}$ is a smooth function and $h(x)$ is the box constraint indicator,*

$$
h(x) = \begin{cases} 0 & \text{if } l_i \leq x_i \leq u_i \text{ for all } i \in \{1, \ldots, n\} \\ \infty & \text{otherwise} \end{cases}
$$

*Assume that $F$ satisfies the proximal-PL condition in the 2-norm with constant constant $\mu_2$ and that $f$ is 2-coordinate-wise Lipschitz in the 2-norm. Then, minimizing*

$$
\min_{x \in \mathbb{R}^n} f(x),
$$
$$
\text{subject to } \langle x, 1 \rangle = \gamma, \ x_i \in [l_i, u_i] \tag{38}
$$

*using 2-coordinate descent with coordinate blocks selected according to the GS-q rule obtains the*
*following linear rate of convergence:*

$$f(x^k) - f^* \leq \left( 1 - \frac{\mu_2}{L_2(n-1)} \right)^k \left( f(x^0) - f^* \right). \tag{39}$$

We provide the proof in Appendix D.2. The proof instantiates a more general result which holds for arbitrary functions $h$ and larger blocks sizes.

## D.1 Proof of Lemma D.3

*Proof.* The proof extends Bertsekas [1998, Proposition 9.22]. Consider $x \in S$. If $\text{supp}(x) = \tau$, then let $d_1 = x$ and we are done. Otherwise, by Lemma D.6 there exists an elementary vector $d_1 \in S$ that is conformal to $x$. Let

$$\gamma = \max \left\{ \gamma \,\middle|\, [x]_j - \gamma[d_1]_j \geq 0 \quad \forall j \text{ with } [x]_j > 0 \quad \text{and} \right.$$

$$\left. [x]_j - \gamma[d_1]_j \leq 0 \quad \forall j \text{ with } [x]_j < 0. \right\}.$$

The vector $\gamma d_1$ is conformal to $x$. Let $\bar{x} = x - \gamma d_1$. If $\text{supp}(x_1) \leq \tau$, choose $d_2 = \bar{x}$ and we are done. Note that $d_2 \in S$ since $S$ is closed under subtraction. Otherwise, let $x = \bar{x}$ and repeat the process. Let $s$ be the number of times this process is conducted. Each iteration reduces the number of non-zero coordinates of $x$ by at least one. Since it terminates when $\text{supp}(x) = \tau$, we have $s \leq n - \tau$.
□

## D.2 Proof of Theorem D.4

We prove the result by instantiating a more general convergence theorem for optimization with linear constraints $Ax = c$, where $A \in \mathbb{R}^{m \times n}$, and general non-smooth regularizers $h$. We assume $A$ is full row-rank and that the proximal operator for $h$ is easily computed. Note that, in this setting, block coordinate descent must operates on blocks $b_i \subset [n]$ of size $m + 1 \leq \tau \leq n$ in order to maintain feasibility of the iterates. Let $U_{b_i}(d_{b_i})$ map block update vector $d_{b_i}$ from $\mathbb{R}^\tau$ to $\mathbb{R}^n$ by augmenting it with zeros and define

$$S_{b_i} = \{ d_{b_i} : AU_{b_i}(d_{b_i}) = 0 \}.$$

That is, $S_{b_i}$ is the null space of $A$ overlapping with block $b_i$.

As mentioned before, the notions of conformal and elementary vectors introduced in the previous section provide necessary tools for our convergence proof. The following Lemmas provide the main show the utility of these definitions for optimization.

**Lemma D.5** (Necoara and Patrascu [2014, Lemma 2])**.** *Given $d \in Null(A)$, if $d$ is an elementary vector of $Null(A)$, then*

$$supp(d) \leq rank(A) + 1.$$

**Lemma D.6** (Bertsekas [1998, Proposition 9.22])**.** *Let $S$ be a subspace of $\mathbb{R}^n$. Then vector $d \in S$ is either a elementary vector of $S$, or there exists an elementary vector $d' \in S$ that is conformal to $d$.*

**Lemma D.7** (Tseng and Yun [2009, Lemma 6.1])**.** *Let $h$ be a coordinate-wise separable and convex function. For any $x$, $x + d \in dom(h)$, let $d$ be expressed as $d = d_1 + \cdots + d_s$ for some $s \geq 1$ and some non-zero $d_t \in \mathbb{R}^n$ conformal to $d$ for $t = 1, \ldots, s$. Then*

$$h(x+d) - h(x) \geq \sum_{t=1}^{s} \left( h\left(x + d_t\right) - h(x) \right).$$

We are now ready to prove our general convergence result for block-coordinate descent with linear constraints and the GS-q block selection rule. We emphasize that in the following theorem: (i) $h$ need not be the indicator for box constraints; (ii) $A$ many consist of many coupling constraints; and (iii) the convergence rate improves with block-size $\tau$, unlike many similar results.

**Proposition D.8.** *Fix block size $\tau \geq m + 1$ and let $\mathcal{B}$ be the set of all blocks $b_i \subset [n]$ of size $\tau$. Consider solving the linearly constrained problem*

$$\min_{x \in \mathbb{R}^n} F(x) := f(x) + h(x),$$

$$\text{subject to } Ax = c$$

*where the gradient of $f$ is $\tau$-coordinate Lipschitz with constant $L_2$ and $h$ is convex and coordinate-wise separable. Suppose $F$ satisfies the proximal-PL inequality in the 2-norm with constant $\mu_2$. Then the block-coordinate descent method with blocks given by the GS-q rule converges as*

$$F(x^k) - F^* \leq \left(1 - \frac{\mu_2}{L_2(n - \tau + 1)}\right)^k \left(F(x^0) - F^*\right).$$

*Proof.* Block-coordinate Lipschitz continuity of $\nabla f$ give the following version of the descent lemma:

$$f(x^{k+1}) \leq f(x^k) + \langle \nabla f(x^k), x^{k+1} - x_k \rangle + \frac{L_2}{2} \|x^{k+1} - x^k\|_2^2$$

We have $x^{k+1} = x^k + U_{b^k}(d^*_{b^k})$ by definition of the update rule. Substituting this into the descent lemma gives

$$f(x^{k+1}) \leq f(x^k) + \langle \nabla_{b^k} f(x^k), d^*_{b^k} \rangle + \frac{L_2}{2} \|d^*_{b_i}\|_2^2$$

$$\Rightarrow f(x^{k+1}) + h(x^{k+1}) \leq f(x^k) + \langle \nabla_{b^k} f(x^k), d_{b^k} \rangle + \frac{L_2}{2} \|d^*_{b_i}\|_2^2 + h(x^{k+1}) + h(x^k) - h(x^k)$$

$$\Rightarrow F(x^{k+1}) \leq F(x^k) + \langle \nabla_{b^k} f(x^k), d^*_{b^k} \rangle + \frac{L_2}{2} \|d^*_{b_i}\|_2^2 + h_{b^k}(x^k_{b^k} + d^*_{b^k}) - h_{b^k}(x^k_{b^k}).$$

Substituting in the choice of coordinate block $b^k$ according to the GS-q rule and the definition of $d^*_{b^k}$ gives

$$F(x^{k+1}) \leq F(x^k) + \min_{b_i \in B} \left\{ \min_{d_{b_i} \in S_{b_i}} \left\{ \langle \nabla_{b_i} f(x^k), d_{b_i} \rangle + \frac{L_2}{2} \|d^*_{b_i}\|_2^2 \right.\right.$$

$$\left.\left. + h_{b_i}(x^k_{b_i} + d_{b_i}) - h_{b_i}(x^k_{b_i}) \right\} \right\}.$$

For clarity, we define the quadratic upper bound to be the function

$$V(x^k, d_{b_i}) = \langle \nabla_{b_i} f(x^k), d_{b_i} \rangle + \frac{L_2}{2} \|d^*_{b^k}\|_2^2 + h_{b_i}(x^k_{b_i} + d_{b_i}) - h_{b_i}(x^k_{b_i}),$$

which gives

$$F(x^{k+1}) \leq F(x^k) + \min_{b_i \in B} \left\{ \min_{d_{b_i} \in S_{b_i}} \left\{ V(x^k, d_{b_i}) \right\} \right\}. \tag{40}$$

We must control that the right-hand-side of (40) in terms of the full-coordinate minimizer

$$d^* = \arg\min d \in \text{Null}(A) \left\{ \langle \nabla f(x^k), d \rangle + \frac{L_2}{2} \|d\|_2^2 + h(x^k + d) - h(x^k) \right\}.$$

in order to apply the prox-PL inequality. We briefly digress and consider conformal realizations of $d^*$ in order to do so.

By lemma D.3, $d^*$ has a conformal realization

$$d^* = d^*_1 + \cdots + d^*_r + d^*_{r+1},$$

where $r \leq n - \tau$ and $d^*_1, \ldots d^*_r$ are elementary vectors of $\text{Null}(A)$ and $d^*_{r+1} \in \text{Null}(A)$. Lemma D.5 gives $\text{supp}(d^*_l) \leq m + 1$; therefore there exists $b_i \in B$ such that $d^*_l \in S_{b_i}$ for all $l = 1, \ldots, r$. By construction, $\text{supp}(d^*_{r+1}) = \tau$ and so there also exists $b_i \in B$ such that $d^*_{r+1} \in S_{b_i}$. Let $\bar{B} \subseteq B$ be the smallest set of blocks such that

$$\forall l \in \{1, \ldots, r+1\}, \exists b_i \in \bar{B}, \quad d^*_l \in S_{b_i},$$

549    and observe that $|\bar{B}| \leq n - 1$.

550

551    Returning to (40), we can use the fact that the value of $V(x^k, d_j)$ obtained at the minimizing block
552    $b^k \in B$ is less than or equal to the average over the subset of blocks $\bar{B}$:

$$\min_{b_i \in B} \left\{ \min_{d_{b_i} \in S_{b_i}} \left\{ V(x^k, d_{b_i}) \right\} \right\} \leq \frac{1}{|\bar{B}|} \sum_{b_i \in \bar{B}} \min_{d_{b_i} \in S_{b_i}} \left\{ V(x^k, d_{b_i}) \right\}. \tag{41}$$

553    Combining this result with (40) and (41), we obtain

$$F(x^{k+1}) \leq F(x^k) + \frac{1}{|\bar{B}|} \sum_{b_i \in \bar{B}} \min_{d_{b_i} \in S_{b_i}} \left\{ V(x^k, d_{b_i}) \right\}$$

$$= F(x^k) + \frac{1}{|\bar{B}|} \min_{d_{b_i} \in S_{b_i}, \forall b_i \in \bar{B}} \left\{ \sum_{b_i \in \bar{B}} V(x^k, d_{b_i}) \right\}$$

$$= F(x^k) + \frac{1}{|\bar{B}|} \min_{d_{b_i} \in S_{b_i}, \forall b_i \in \bar{B}} \left\{ \langle \nabla f(x^k), \sum_{b_i \in \bar{B}} d_{b_i} \rangle + \sum_{b_i \in \bar{B}} \frac{L_2}{2} \|d_{b_i}\|^2 \right.$$

$$\left. + \sum_{b_i \in \bar{B}} \left( h_{b_i}(x^k_{b_i} + d_{b_i}) - h_{b_i}(x^k_{b_i}) \right) \right\}. \tag{42}$$

For all $b_i \in \bar{B}$, substituting any $d_{b_i} \in S_{b_i}$ for the vector in $S_{b_i}$ that minimizes (42) can only increase
554    the upper bound. Choosing the $d_l^*$ corresponding to each block $b_i \in \bar{B}$ yields

$$\leq F(x^k) + \frac{1}{|\bar{B}|} \left( \langle \nabla f(x^k), \sum_{l=1}^{r+1} d_l^* \rangle + \sum_{l=1}^{r+1} \frac{L_2}{2} \|d_l^*\|^2 \right.$$

$$\left. + \sum_{l=1}^{r+1} \left( h_{b_i}(x^k_{b_i} + d_l^*) - h_{b_i}(x^k_{b_i}) \right) \right).$$

555    We now use $d^* = \sum_{l=1}^{r+1} d_l^*$ and apply lemma D.7 twice to obtain

$$F(x^{k+1}) \leq F(x^k) + \frac{1}{|\bar{B}|} \left( \langle \nabla f(x^k), \sum_{l=1}^{r+1} d_l^* \rangle + \frac{L_2}{2} \|d^*\|^2 \right.$$

$$\left. + \sum_{l=1}^{r+1} \left( h_{b_i}(x^k_{b_i} + d_l^*) - h_{b_i}(x^k_{b_i}) \right) \right)$$

$$F(x^{k+1}) \leq F(x^k) + \frac{1}{|\bar{B}|} \left\{ \langle \nabla f(x^k), d^* \rangle + \frac{L_2}{2} \|d^*\|^2 + h(x^k + d^*) - h(x^k) \right\}$$

$$= F(x^k) + \frac{1}{|\bar{B}|} \min_{d \in S} \left\{ \langle \nabla f(x^k), d \rangle + \frac{L_2}{2} \|d\|_2^2 + h(x^k + d) - h(x^k) \right\}. \tag{43}$$

556    Applying the prox-PL inequality in the $\|\cdot\|_2$ norm gives

$$F(x^{k+1}) \leq F(x^k) - \frac{\mu_2}{|\bar{B}|}(F(x^k) - F^*)$$

$$= F(x^k) - \frac{\mu_2}{L_2(n - \tau + 1)}(F(x^k) - F^*).$$

557    Subtracting $F^*$ from both sides and applying the inequality recursively completes the proof.    □

558    Instantiating Proposition D.8 with $A = 1^\top$, $c = \gamma$, $\tau = 2$ and

$$h(x) = \begin{cases} 0 & \text{if } l_i \leq x_i \leq u_i \text{ for all } i \in \{1, \ldots, n\} \\ \infty & \text{otherwise} \end{cases}$$

559    is sufficient to obtain Theorem D.4.

 # E   Greedy Rules Depending on Coordinate-Wise Constants

561 We first derive the greedy GS-q rule, then steepest descent in the L-norm, and then give a dimension-
562 independent convergence rate based on the L-norm.

## E.1   GS-q Rule with Coordinate-Wise Constants

564 The GS-q rule under an equality constraint and coordinate-wise Lipschitz constants is given by

$$\arg\min_{b} \left\{ \min_{d_b | d_i + d_j = 0} \langle \nabla_b f(x), d_b \rangle + \frac{L_i}{2} d_i^2 + \frac{L_j}{2} d_j^2 \right\}. \tag{44}$$

565 **Solving for $d_b$.**   We first fix $b$ and solve for $d_b$. The Lagrangian of the inner minimization in (44) is:

$$\mathcal{L}(d, \lambda) = \langle \nabla_b f(x), d_b \rangle + \frac{L_i}{2} d_i^2 + \frac{L_j}{2} d_j^2 + \lambda(d_i + d_j).$$

566 Set the gradient with respect $d_i$ to zero we get

$$\nabla_i f(x) + L_i d_i + \lambda = 0,$$

567 and solving for $d_i$ gives

$$d_i = \frac{-\nabla_i f(x) - \lambda}{L_i}. \tag{45}$$

568 Similarly, we have

$$d_j = \frac{-\nabla_j f(x) - \lambda}{L_j}. \tag{46}$$

569 Since $d_i = -d_j$ we have

$$\frac{-\nabla_i f(x) - \lambda}{L_i} = \frac{\nabla_j f(x) + \lambda}{L_j},$$

570 and solving for $\lambda$ gives

$$\lambda = \frac{-(L_j \nabla_i f(x) + L_i \nabla_j f(x))}{L_i + L_j}. \tag{47}$$

571 Substituting (47) in (45) gives

$$\begin{aligned}
d_i &= \frac{1}{L_i} \left( -\nabla_i f(x) - \frac{-(L_j \nabla_i f(x) + L_i \nabla_j f(x))}{L_i + L_j} \right) \\
&= \frac{1}{L_i} \left( \frac{-L_i \nabla_i f(x) - L_j \nabla_i f(x) + L_j \nabla_i f(x) + L_i \nabla_j f(x)}{L_i + L_j} \right) \\
&= \frac{1}{L_i} \left( \frac{-L_i \nabla_i f(x) + L_i \nabla_j f(x)}{L_i + L_j} \right) \\
&= -\frac{\nabla_i f(x) - \nabla_j f(x)}{L_i + L_j},
\end{aligned}$$

572 and similarly

$$d_j = \frac{\nabla_i f(x) - \nabla_j f(x)}{L_i + L_j}.$$

573 **Solving for $b$.** We now use the optimal $d_i$ and $d_j$ in (44),

$$\arg\min_b \left\{ \nabla_i f(x) d_i + \nabla_j f(x) d_j + \frac{L_i}{2} d_i^2 + \frac{L_j}{2} d_j^2 \right\}$$

$$\equiv \arg\min_b \left\{ \nabla_i f(x) d_i - \nabla_j f(x) d_i + \frac{L_i}{2} d_i^2 + \frac{L_j}{2} d_i^2 \right\}$$

$$\equiv \arg\min_b \left\{ (\nabla_i f(x) - \nabla_j f(x)) d_i + \frac{L_i + L_j}{2} d_i^2 \right\}$$

$$\equiv \arg\min_b \left\{ -\frac{(\nabla_i f(x) - \nabla_j f(x))^2}{L_i + L_j} + \frac{(\nabla_i f(x) - \nabla_j f(x))^2}{2(L_i + L_j)} \right\}$$

$$\equiv \arg\min_b \left\{ -\frac{1}{2} \frac{(\nabla_i f(x) - \nabla_j f(x))^2}{L_i + L_j} \right\}$$

$$\equiv \arg\max_b \left\{ \frac{(\nabla_i f(x) - \nabla_j f(x))^2}{L_i + L_j} \right\}.$$

## 574 E.2 Steepest Descent with Coordinate-Wise Constants

575 Here, we show that steepest descent in the $L$-norm always admits at least one solution which updates
576 only two coordinates. Steepest descent in the $L$-norm, subject to the equality constraint, takes steps
577 in the direction $d$ that minimizes the following model of the objective:

$$d \in \arg\min_{d \in \mathbb{R}^n | d^T 1 = 0} \left\{ \nabla f(x)^T d + \frac{1}{2\alpha} ||d||_L^2 \right\}, \tag{48}$$

578 This is a convex optimization problem for which strong duality holds. Introducing a dual variable
579 $\lambda \in \mathbb{R}$, we obtain the Lagrangian

$$\mathcal{L}(d, \lambda) = \nabla f(x)^T d + \frac{1}{2\alpha} ||d||_L^2 - \lambda(d^T 1).$$

580 The subdifferential with respect to $d$ and $\lambda$ yields necessary and sufficient optimality conditions for a
581 steepest descent direction,

$$\nabla_d \mathcal{L}(d, \lambda) = \nabla f(x) + \frac{1}{2\alpha} g - \lambda 1 = 0$$

$$\text{(for some subgradient } g \in \partial ||d||_L^2)$$

$$\nabla_\lambda \mathcal{L}(d, \lambda) = d^T 1 = 0.$$

582 The second condition is simply feasibility of $d$, while from the first we obtain,

$$2\alpha(-\nabla f(x) + \lambda 1) \in \partial ||d||_L^2$$

$$\alpha(-\nabla f(x) + \lambda 1) \in ||d||_L (\sqrt{L} \odot \text{sgn}(d)), \tag{49}$$

583 where element $m$ of $\text{sgn}(d)$ is 1 if $d_m$ is positive, $-1$ if $d_m$ is negative, and can be any value in $[-1, 1]$
584 if $d_m$ is 0. The following lemma shows that these conditions are always satisfied by a two-coordinate
585 update.

586 **Lemma E.1.** *Let $\alpha > 0$. Then at least one steepest descent direction with respect to the 1-norm has*
587 *exactly two non-zero coordinates. That is,*

$$\min_{d \in \mathbb{R}^n | d^T 1 = 0} \nabla f(x)^T d + \frac{1}{2\alpha} ||d||_L^2 =$$

$$\min_{i,j} \left\{ \min_{d_{ij} \in \mathbb{R}^2 | d_i + d_j = 0} \nabla_{ij} f(x)^T d_{ij} + \frac{1}{2\alpha} ||d_{ij}||_L^2 \right\}. \tag{50}$$

588 *Proof.* Similar to the steepest descent in the 1-norm, the proof follows by constructing a solution to
589 the steepest descent problem in Eq. 48 which only has two non-zero entries. Let $i$ and $j$ maximize

$(\nabla_i f(x) - \nabla_j f(x))/(\sqrt{L_i} + \sqrt{L_j})^2$. Our proposed solution is $d$ such that $d_i = -\delta, d_j = \delta$ for some $\delta \in \mathbb{R}$ and $d_{k,k \neq i,j} = 0$. In order for this relationship in (49) to hold, we would require

$$-\alpha \nabla f(x) + \alpha \lambda 1 \in \|d\|_L (\sqrt{L} \odot sgn(d)). \tag{51}$$

From the definition of L-norm and our definition of $d$ that

$$\|d\|_L = \sqrt{L_i}\delta + \sqrt{L_j}\delta$$
$$= \delta(\sqrt{L_i} + \sqrt{L_j}).$$

Also, we know that $sgn(d_i) = -1$, $sgn(d_j) = 1$, $sgn(d_k) = [-1, 1]$. Therefore, we would need

$$-\alpha \nabla_i f(x) + \alpha \lambda = -\delta \sqrt{L_i}(\sqrt{L_i} + \sqrt{L_j}) \tag{52}$$
$$-\alpha \nabla_j f(x) + \alpha \lambda = \delta \sqrt{L_j}(\sqrt{L_i} + \sqrt{L_j}) \tag{53}$$
$$-\alpha \nabla_k f(x) + \alpha \lambda = \delta \sqrt{L_k}(\sqrt{L_i} + \sqrt{L_j})[-1, 1] \tag{54}$$

From (52), $\lambda = \nabla_i f(x) - \frac{\delta}{\alpha} \sqrt{L_i}(\sqrt{L_i} + \sqrt{L_j})$. Substituting $\lambda$ in (53), we get

$$-\alpha \nabla_j f(x) + \alpha \nabla_i f(x) - \delta \sqrt{L_i}(\sqrt{L_i} + \sqrt{L_j}) = \delta \sqrt{L_j}(\sqrt{L_i} + \sqrt{L_j})$$
$$\alpha \nabla_i f(x) - \alpha \nabla_j f(x) = \delta(\sqrt{L_i} + \sqrt{L_j})(\sqrt{L_i} + \sqrt{L_j}),$$

From this we get,

$$\delta = \frac{\alpha}{(\sqrt{L_i} + \sqrt{L_j})^2}(\nabla_i f(x) - \nabla_j f(x)). \tag{55}$$

Using $\lambda$ in (54) means that for variables $k \neq i$ and $k \neq j$ that we require

$$-\alpha \nabla_k f(x) + \alpha \nabla_i f(x) - \delta \sqrt{L_i}(\sqrt{L_i} + \sqrt{L_j}) \in \delta \sqrt{L_k}(\sqrt{L_i} + \sqrt{L_j})[-1, 1]$$
$$-\alpha(\nabla_i f(x) - \nabla_k f(x)) \in \delta(\sqrt{L_i} + \sqrt{L_k})(\sqrt{L_i} + \sqrt{L_j})[-1, 1]$$
$$-\alpha \frac{\nabla_k f(x) - \nabla_i f(x)}{(\sqrt{L_i} + \sqrt{L_k})} \in \delta(\sqrt{L_i} + \sqrt{L_j})[-1, 1]$$

Using the definition of $\delta$ (55) this is equivalent to

$$-\frac{\nabla_i f(x) - \nabla_k f(x)}{\sqrt{L_i} + \sqrt{L_k}} \in \frac{\nabla_i f(x) - \nabla_j f(x)}{\sqrt{L_i} + \sqrt{L_j}}[-1, 1],$$

which holds due to the way we chose $i$ and $j$.

We have shown that a two-coordinate update $d$ satisfies the sufficient conditions to be a steepest descent direction in the $L$-norm. $\square$

### E.3 Convergence result for coordinate-wise Lipschitz case

Lemma E.1 allows us to give a dimension-independent convergence rate of a greedy 2-coordinate method that incorporates the coordinate-wise Lipschitz constants, by relating the progress of the 2-coordinate update to the progress made by a full-coordinate steepest descent step. If we use $L_L$ as the Lipschitz-smoothness constant in the $L$-norm, then by the descent lemma we have

$$f(x^{k+1}) \leq f(x^k) + \nabla f(x^k)^T d^k + \frac{L_L}{2}\|d^k\|_L^2.$$

From Lemma E.1, if we use the greedy two-coordinate update to set $d^k$ and use a step size of $\alpha = 1/L_L$ we have

$$f(x^{k+1}) \leq f(x^k) + \min_{d|d^T 1 = 0}\left\{\nabla f(x^k)^T d + \frac{L_L}{2}\|d\|_L^2\right\}.$$

Now subtracting $f^*$ from both sides and the proximal-PL assumption in the $L$-norm,

$$f(x^{k+1}) - f(x^*) \leq f(x^k) - f(x^*) - \frac{1}{2L_L}\mathcal{D}(x^k, L_L)$$

$$= f(x^k) - f(x^*) - \frac{\mu_L}{L_L}(f(x^k) - f^*)$$

$$= \left(1 - \frac{\mu_L}{L_L}\right)(f(x^k) - f^*)$$

It is possible to obtain a faster rate using a smallest setting of the $L_i$ such that $f$ is 1-Lipschitz in the $L$-norm. However, it is not obvious how to find such $L_i$ in practice.

# F   General Equality Constraints

Rather a constraint of the form $\sum_i x_i = \gamma$, we could also consider general equality constraints of the form $\sum_i a_i x_i = \gamma$ for positive weights $a_i$. In this case the greedy rule would be

$$\arg\max_{i,j}\left\{\frac{a_j\nabla_i f(x) - a_i\nabla_j f(x)}{a_1 + a_2}\right\},$$

and we could use a $\delta^k$ of the form

$$\delta^k = -\frac{\alpha}{a_1 + a_2}[a_2\nabla_1 f(w^k) - a_1\nabla_2 f(w^k)].$$

Unfortunately, the greedy rule in this case appears to requirer $O(n^2)$. However, if re-parameterized in terms of variables $x_i/a_i$ then the constraint is transformed to $\sum_i x_i = \gamma$ and we can use the methods discussed in this work (although the ratio approximation also relies on re-parameterization so makes less sense here).

We could also consider the case performing greedy coordinate descent methods with a set of linear equality constraints. With $m$ constraints, we expect this to require updating $m + 1$ variables. Although it is straightforward to define greedy rules for this setting, it is not obvious that they could be implemented efficiently.

# G   Additional Experiments

In Figure 3, we repeat the scaled version of our equality-constrained experiment with different seeds. We updated the Greedy(Ratio) method with

$$i_k \in \arg\max_i(\nabla_i f(x^k) - \mu)/\sqrt{L_i}, \quad j_k \in \arg\min_j(\nabla_j f(x^k) - \mu)s/\sqrt{L_j}, \tag{56}$$

where $\mu$ is the mean of $\nabla f(x^k)$. We observed that the Greedy(Ratio) and Greedy(Switch) approximations consistently performed similarly to the exact Greedy Li method.

We repeated the experiment that compares different greedy methods under equality and bound constraints with different seeds in Figures 4, 5, and 6. We see that the GS-q and GS-1 have a small but consistent advantage in terms of decreasing the objective while the GS-s and GS-1 rules have a consistent advantage in terms of moving variables to the boundaries. Finally, we see that the GS-1 rule only updates 2 variables on most iterations (over 85%) while it updates 3 or fewer variables on all but a few iterations.

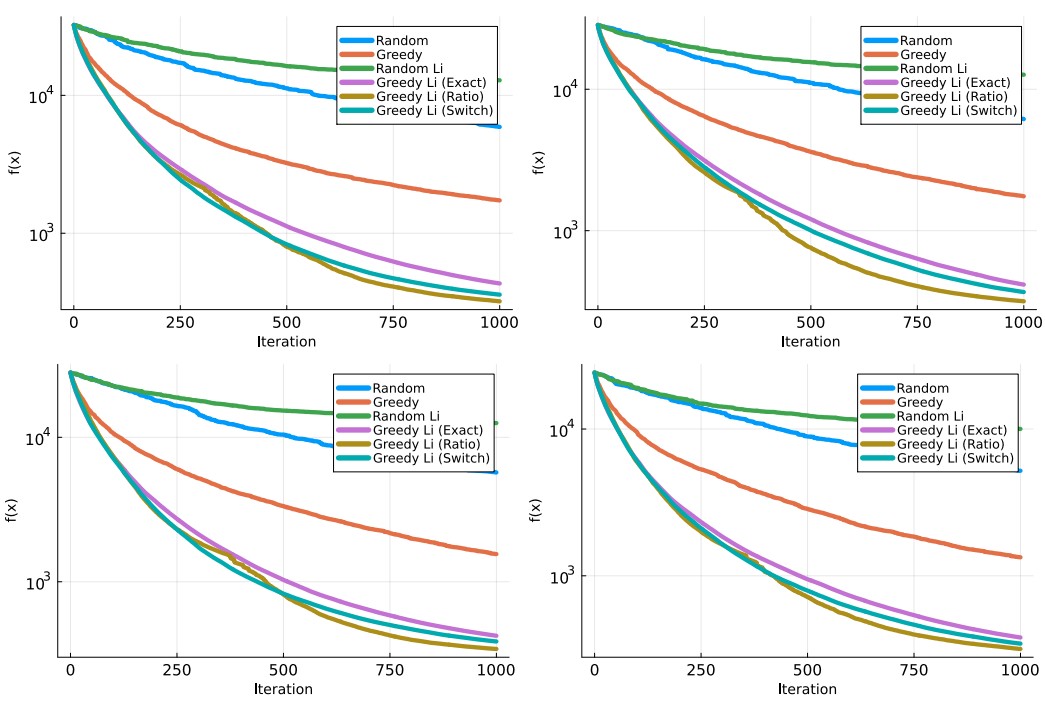

Figure 3: Comparison of different random and greedy rules under 4 choices for the random seed used to generate the data (and for the sampling in the randomized methods).

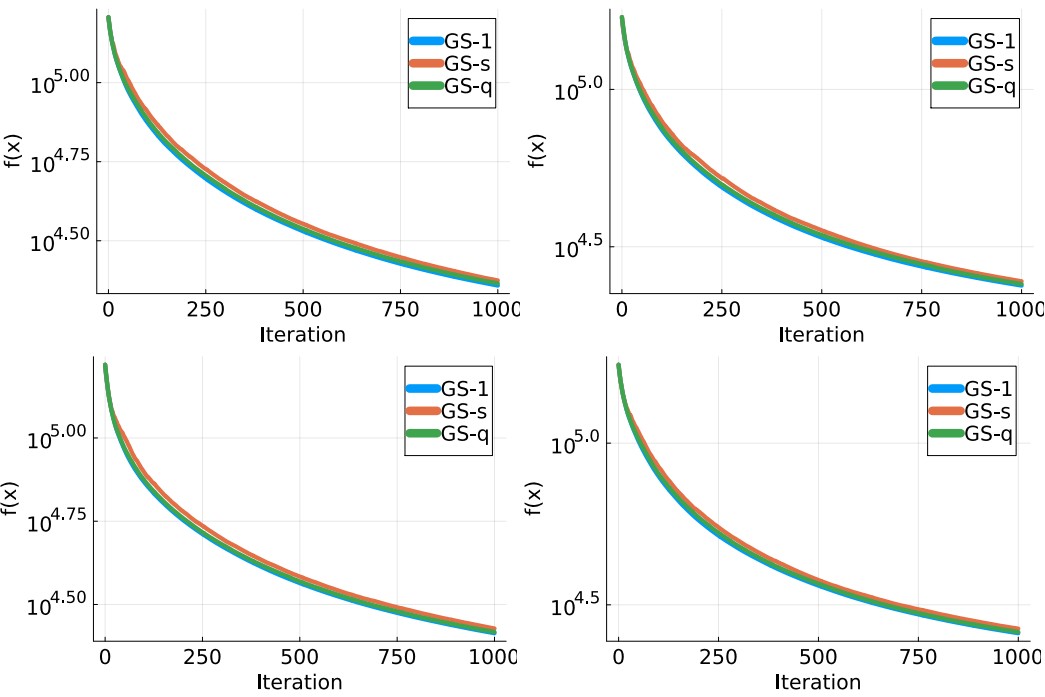

Figure 4: Comparison of different greedy rules under 4 choices for the random seed used to generate the data.

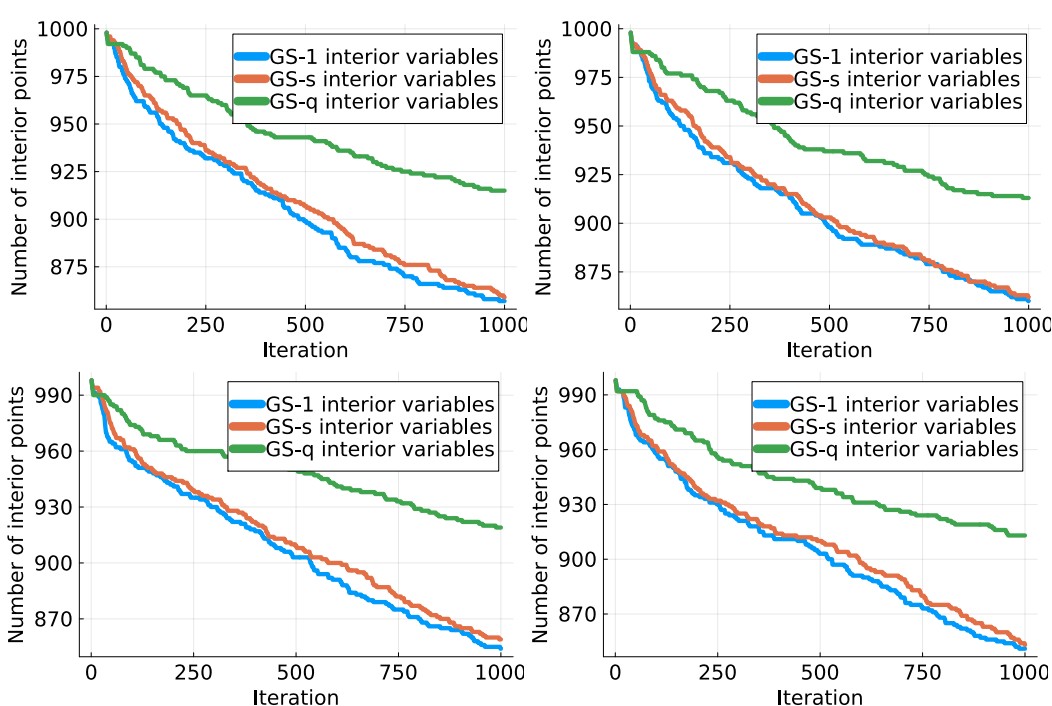

Figure 5: Comparison of number of interior variables updated by GS-1, GS-q and GS-s in every iteration for data generated by different random seed

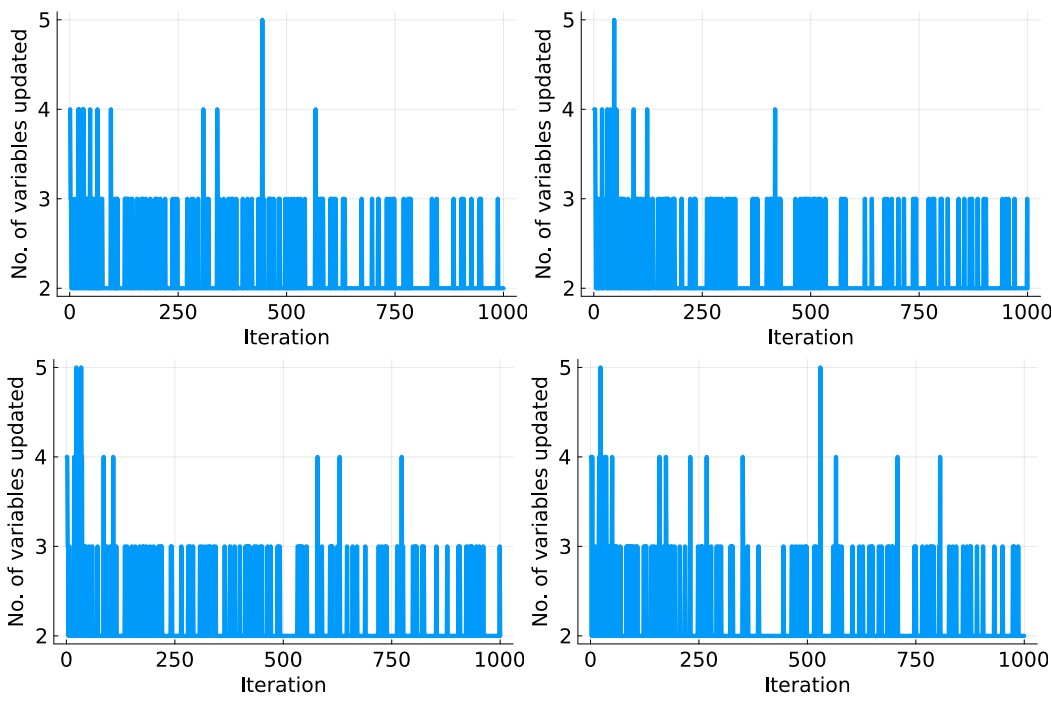

Figure 6: Number of variables updated by GS-1 with different random seed used to generate the data.