# OpenReview forum: "Analyzing and Improving Greedy 2-Coordinate Updates For Equality-Constrained Optimization via Steepest Descent in the 1-Norm"
_NeurIPS.cc/2023/Conference — Submitted to NeurIPS 2023_

### Official Review · Reviewer_pHfj · 2023-06-23

**Soundness:** 3 good
**Presentation:** 3 good
**Contribution:** 3 good
**Rating:** 5
**Confidence:** 4

**Summary:**

The paper presents new update rules for block-coordinate descent (BCD) methods to minimize a smooth function subject to one linear equality constraint (precisely, all variables must sum to 1) and possibly a box constraint. A popular method to solve large-scale problems of this type is BCD with blocks of size 2 (i.e. we always update 2 coordinates at a time). A prominent example is SVM training in LibSVM.

For problems without the box constraint, the key observation is lemma 2.1, which shows that greedy version of such updates is equivalent to 1-norm steepest descent. This allows the authors to employ known results from non-smooth optimization and propose better update rules and derive their convergence rate. This rate is better than known rates for non-greedy updates.

For problems with a box constraint, the situation is more complicated because Lemma 2.1 does not apply. The authors nevertheless propose a new update (GS-1) based on 1-norm steepest descent, which is not guaranteed to change at most 2 variables in every update, but more than 2 variables are changed only in a finite number of updates. This can be seen as a greedy version of BCD with small blocks. The update has a better convergence rate than known updates that can be computed in reasonable (less than $O(n^2)$) time.

The theory is supported by a synthetic toy experiment on the linear LS problem with one linear equality constraint and a box constraint.

**Strengths:**

Crisp and elegant theoretical result, non-trivial.
However, let me admit that convergence analysis of optimization algorithms is not precisely my expertise, so I am not reliably judge the novelty (these or similar results may be already known or obvious, ...).

The text is precise and clear.

**Weaknesses:**

A major weakness is experiments: they are limited only to a simple problem (linear least squares) on synthetic data, end even then the data variability tested is small (only square system). To my understanding, the GS-1 update has never been applied to SVM training - so such experiments would be very interesting.

Moreover, in the experiments the difference $f(x)-f^*$ (vertical axis in Figure 1) is never shown to converge to zero, it is shown only in the range $10^{4.8} - 10^{5.2}$. In my experience, it sometimes happens that greedy updates improve the objective better initially but later they slow down and cyclic/random updates catch up. Or, perhaps, the vertical axis in Figure 1 shows $f(x)$ rather than $f(x)-f^*$ (as in the Supplement)?

Minor issues in the text:

- The five-line derivation on line 79-80 is trivial - it can be shortened to save space.

- Lemma 2.1 is very much related to the well-known result that greedy coordinate descent is equivalent to 1-norm steepest descent (see, e.g., Section 9.4 in Boyd's book on convex optimization). This may deserve a citation.

- Lemma 2.1 would be clearer if $\nabla f(x)$ were replaced by a general vector, say c.

- Line 128: mentioning SVM is not OK here because SVM needs also a box constraint.

- Line 139: the notion of "dimension independent convergence rate" is not well-defined because the problem dimension can be hidden in constants (such as $L_1, \mu_1$). But I understand that this notion may be widely used.

**Questions:**

How would Figure 1 look like if the iterations continued to a very small (10^-6) error f(x)-f^* ?

How does the GS-1 update behave in SVM training?

Are there any other problem types on which you tested the new updates? Were there any negative results (i.e., the convergence was slow in practice despite theoretical guarantees)?

**Limitations:**

The experiments are limited to a very simple problem with synthetic data.

Though the theoretical results are elegant, the optimization problem considered has a rather limited applicability.

---

> ### Author Rebuttal · Authors · 2023-08-09
>
> Thank you for pointing out these minor issues. We will fix them.
>
> > "How does the GS-1 update behave in SVM training?"
>
> Please see the discussion of experiments in the general response.
>
> > "How would Figure 1 look like if the iterations continued to a very small (10^-6) error f(x)-f^*?"
>
> Good question. In the PDF attached to the general response we show the curves if the methods are run much longer. We see that the greedy methods find a reasonable solution far before random methods.
>
> If we consider SVMs where we also have bound constraints, then the difference between random and greedy tends to be even larger. This is because greedy methods eventually only focus on updating the support vectors; greedy methods speed up at this point since they are solving a lower-dimensional problem. In contrast, random updates continue to select the non-support vectors which eventually do not move (so most random iterations do nothing). Taken together with the similar iteration cost and slower rate, random methods are not as popular as greedy methods for training SVMs.
>
> > "Were there any negative results?"
>
> Since the time of submission, we found that it is possible for pure random to outperform pure greedy if we are using the Li values to set the step size and if the Li values are very different. However, we always found that greedy was faster than any form of random update if the Li values are similar or if we use the Li values in the greedy update.

---

> > ### Comment · Reviewer_pHfj · 2023-08-15
> >
> > I thank the authors for their rebuttal. However, the PDF with additional plots (added to the general rebuttal) does not convincingly support the authors' claim that "the greedy methods find a reasonable solution far before random methods". In Figure 2 of the PDF, the "Random" method (blue line) is only slightly worse than the greedy methods, and it is so only for initial $\approx 10^4$ iterations. Since it is not clear at which accuracy the algorithm is supposed to be terminated, the fact that some method is better than the others for some initial number of iterations might be irrelevant.
> >
> > It is a pity that the authors did not show also the plots zoomed in for small values of $f(x)-f^*$ because now it is hard to see how the methods compare in this range. (this applies both to Figure 1 and 2)
> >
> > Is Figure 2 for the real data (SVM) or the synthetic data?

---

> > > ### Author Response · Authors · 2023-08-17
> > >
> > > At the final iteration on the plot the 3 greedy methods incorporating the Li values are at numerical precision, the classic greedy method has a sub-optimality close to 0, while the random method has a sub-optimality around 30. It is true that we should have included a zoomed-in version of the end or plotted on a log-scale (which we will update in the paper).
> > >
> > > (We made Figure 2 in response to the comment "How would Figure 1 look like if the iterations continued to a very small (10^-6) error f(x)-f^", so it is on the same synthetic data. )

---

### Official Review · Reviewer_Ffqu · 2023-07-06

**Soundness:** 4 excellent
**Presentation:** 4 excellent
**Contribution:** 3 good
**Rating:** 7
**Confidence:** 3

**Summary:**

The first goal of the paper is to minimize a smooth function subject to a summation constraint. The authors demonstrate that the greedy 2-coordinate descent (CD) method, when applied to the problem with equality constraints, achieves a linear rate of convergence under the proximal PL inequality under the L1-norm formulation. Notably, this convergence rate remains unaffected by the problem dimension, which sets it apart from random selection methods. Furthermore, they establish that there exists at least one steepest descent direction with respect to L1-norm, which can be utilized as a 2-coordinate descent update. They leverage this relationship to derive the convergence rates of the CD algorithm.
Additionally, they explore the minimization involving a summation constraint and prescribed lower and upper bounds on the coordinates. They demonstrate that employing bound- and summation-constrained steepest descent in the L1-norm guarantees significant progress at each step, unlike the GS-s rule. Moreover, this method (called GS-1) is computationally more efficient than GS-q, requiring only $O(n \log(n))$ iterations instead of $O(n^2)$

**Strengths:**

1. The paper gives linear convergence rates for greedy 2-coordinates CD, and the steepest descent in l1-norm under proximal PL for l1-norm. For problems with equality constraints, this work shows that greedy methods may be faster by a factor ranging from $O(n)$ up to $O(n^2)$ than methods picking coordinates at random,

2. For problems with bound constraints and equality constraints, the paper shows that GS-1 rule has the benefits of both GS-q and GS-s. Contrary to GS-q, GS-1 rule can be implemented in $O(n \log n)$ and contrary to GS-s, GS-1 guarantees non-trivial progress
at each iteration.

3. The authors also proved a linear convergence rate for GS-q rule under proximal PL for L2-norm. It is not straightforward to use prox-PL to obtain the linear convergence rate for this update rule. They consider the conformal realization used by Tseng and Yun [2009] to upper bound the extra terms in descent inequality by a notion of gradient mapping used in prox-PL’s definition.

4. The paper is clear and well written.

**Weaknesses:**

The experimental part is limited, which is fine for a theoretical optimization paper, but for a ML venue and given the claim made in the paper that many problems in ML require to satisfy an equality constraint, such as discrete probabilities or SVMs with an unregularized bias term, one could expect the experimental section to show the power of GS-1 for other such examples than only a synthetic equality-constrained least square problem.

**Questions:**

1. L42-44: More comments are needed when the authors write that ``despite LIBSVM being perhaps the most widely-used CD method of all time, current analyses of greedy 2-coordinate updates either result in sublinear convergence rates or do not lead to faster rates than random selection [Tseng and Yun, 2009, Beck, 2014]." Did they use proximal PL inequality in L1-norm to analyze the convergence rate of greedy 2-coordinate methods? The linear convergence rate of this method may be due to the PL property.

2. L47-49 "The analysis is based on an equivalence between the greedy update and equality-constrained steepest descent in the 1-norm. This leads to a simple dimension-independent analysis of greedy selection showing that it can converge substantially faster than random selection." Under which assumptions is the faster convergence proven?

3. L. 66 ``If f is continuous, this update is guaranteed to decrease $f$ for sufficiently small $\alpha_k$'' Please quote a relevant paper, or a short proof for this sentence.

4. Question 4 removed as workshop papers are not listed in reviews.

5. L91 “For lower bounding sub-optimality in terms of the 1-norm, we introduce the proximal-PL inequality in the 1-norm.” What do you mean by lower bounding sub-optimality? As far as I can see, you are trying to give an upper bound on sub-optimality in Theorem 2.3.

6. Definition 2.2. “A function, that is L1-Lipschitz”:  is it L1-Lipschitz gradient? Did this definition not appear in [Karimi et al. 2016]?

7. Does $f=g(Ax)$ satisfy proximal PL condition in the L1-norm for strongly convex function $f$ and a singular matrix $A$? Can we verify this property for functions in practice such as the SVM dual problem?

8. Did [She and Schmidt 2017] assume the same PL condition in the L1-norm (or L2-norm)?

9. Connection between the prox-PL function used in Theorem 3.2 and prox-PL as defined only by (9) in Def. 2.2: could you give an example of a function that satisfies this new assumption?

10. Some minor questions/typos in the appendix:

L372 Should $d_1 + d_2$ not be $d_i + d_j$

L378 second line before end: one $)$ too much

L390 Why do you have $\lambda 1$ in (29) and not simply $\lambda$

L428 $d_1 + d_2 + d_3$ should be $d_1 + d_2 - d_3$

L437 Why do we have the parameter $x^\prime$Conv$(\{x,x+d\})$~?

----
The authors have addressed adequately my questions; they intend to conduct extensive experiments. This is indeed missing in the current version of the paper, it is difficult to evaluate that part with the paper as is. But I find the theoretical contribution solid enough to justify publication.

---

> ### Author Rebuttal · Authors · 2023-08-09
>
> Thank you for carefully reading our paper; your comments will help us improve the manuscript. We respond to the highlighted weakness and your questions below.
>
> > "The experimental part is limited"
>
> Please see the discussion of experiments in the general reply.
>
> > 1. "More comments are needed when the authors write ... current analyses of greedy 2-coordinate updates either result in sublinear convergence rates or do not lead to faster rates than random selection [Tseng and Yun, 2009, Beck, 2014]"
>
> Beck considers a very-general class of problems where the proximal-PL condition is not necessarily satisfied. But in this general setting it is only possible to obtain sublinear rates, which would be slower than the linear rates known for SVMs.
>
> Tseng and Yun consider error bounds, which are equivalent to the proximal-PL inequality for Lipschitz-smooth functions [Karimi et al., 2016, Appendix G] and allow linear rates. But Tseng and Yun only show asymptotic rates and measure the error bounds in the 2-norm. We show that keeping the analysis in the 1-norm is the key to obtaining a fast rate.
>
> > 2. Under which assumptions is the faster convergence proven?
>
> It is sometimes hard to compare rates because they can be measured in different ways. This is particularly true among 2-CD papers. However, a clean comparison can be done under the commonly-used assumptions of Theorem 3.2: twice-differentiable $f$, 2-coordinate Lipschitz with constant $L_2$, step size of $1/L_2$, and proximal-PL. In this setting the greedy rate (11) always gains a factor of at least $2n$ over the random rate (12), and may gain a factor as large as $2n^2$.
>
> > 3. "If f is continuous, this update is guaranteed to decrease $f$ for sufficiently small $\alpha_k$."
>
> Here is a quick proof of this fact. Starting from 2-coordinate smoothness of the function $f$ and considering any pair of indices $i \neq j$, we have
> $$
> \begin{aligned}
> f(x + d_{ij})
> &\leq f(x) + \nabla_{ij} f(x)^\top d_{ij} + \frac{1}{L} \|d_{ij}\|_2^2\\\\
> &= f(x) - \frac{\alpha}{2} (1 - \frac{\alpha L}{2}) \left[ (\nabla_i f(x))^2 + (\nabla_j f(x))^2 \right]
>     + \alpha (1 - \frac{\alpha L}{2}) \nabla_i f(x) \nabla_j f(x).
> \end{aligned}
> $$
>
> Take $\alpha < L / 2$. If $\nabla_i f(x) \nabla_j f(x) < 0$, then the proof is done. Otherwise, observe that $2 \nabla_i f(x) \nabla_j f(x) \leq (\nabla_i f(x))^2 + (\nabla_j f(x))^2$ by the AM-GM inequality, with strictness unless $\nabla_i f(x) = \nabla_j f(x)$. Thus, sufficiently small $\alpha$ guarantees progress as long as the two gradients are not equal.
>
> > 4. "similar claim in ... NeurIPS 2022 Workshop"
>
> We note that workshop NeurIPS papers are considered non-archival (see the "NeurIPS 2023 FAQ for Authors").
>
> > 5. "What do you mean by lower bounding sub-optimality?"
>
> This is a good point. The proximal-PL condition should be stated as giving a lower bound on the optimal function value $f^*$ as opposed to the sub-optimality. Specifically, the lower bound would be:
> $$f^* \geq f(x) - \frac{1}{2\mu_1}\mathcal{D}(x,L_1)$$
> This is probably the nicer way to define the proximal-PL inequality as well.
> We will correct this in the paper.
>
> > 6. "is it L1-Lipschitz gradient?"
>
> Yes, agreed.
>
> > 7a. "Does $f= g(AX)$ satisfy proximal PL condition in the L1-norm for strongly convex function f and a singular matrix A?"
>
> Yes. Karimi et al. [2016, Appendix F - Case 3] implies that this problem class satisfies the proximal-PL inequality in the 2-norm (though there appears to be a typo in that result - the Hoffman constant should be squared), and from our Appendix C this implies that it satisfies the proximal-PL inequality in the 1-norm.
>
> > 7b. "Can we verify this property for functions in practice such as the SVM dual problem?"
>
> It is more difficult to verify the proximal-PL and PL conditions than it is to verify conditions like convexity and strong-convexity, since we have fewer rules for composing operations that preserve PL. However, the SVM dual problem is covered by Karimi et al. [2016, Appendix F - Case 3 or Case 4] so it will always satisfy the proximal-PL inequality.
>
> > 8. "Did [She and Schmidt 2017] assume the same PL condition in the L1-norm (or L2-norm)?"
>
> They use the standard proximal-PL condition in the L2-norm.
>
> > 9. "Connection between the prox-PL function used in Theorem 3.2 and prox-PL as defined only by (9) in Def. 2.2: could you give an example of a function that satisfies this new assumption?"
>
> The difference between the two settings is the choice of proximal regularizer. Note that any function which is strongly convex in the L1-norm is prox-PL according to both definitions (we will add a proof of this fact to Appendix C). Additional examples include SVMs (in the dual) and optimizing a strictly-convex function over the probability simplex.
>
> > 10. "minor questions/typos in the appendix"
>
> Will fix these, thanks.

---

> > ### Comment · Reviewer_Ffqu · 2023-08-12
> > **Empirical evaluation**
> >
> > My questions on the theory front are adequately addressed, thank you.
> >
> > About the experimental part: the authors show in Figure 2 in the paper that on the synthetic equality-constrained least square problem, GS-s rule results in the slowest convergence rate. Two of the additional experiments on the SVM real datasets show that GS-s converges as fast as GS-q and GS-1, while the third one shows that GS-s is actually faster than GS-q. Is there any trend (GS-1 being always at least as fast as the GS-q and GS-s) or do all three perform eventually in a similar range, without a clear predictable advantage for either one of them? A more extensive and quantitative empirical evaluation on real datasets would be welcome.

---

> > > ### Author Response · Authors · 2023-08-13
> > >
> > > Thank you for reading and responding with your valuable feedback.
> > > We were surprised by the superior performance of GS-s over GS-q on some real datasets, although in subsequent experiments their performance was very similar, particularly after feature normalization. To comprehensively explore this phenomenon, we intend to conduct an extensive set of experiments, to tease apart various practical issues and to pinpoint instances where empirical behavior differs from the presented theory.

---

### Official Review · Reviewer_yjwM · 2023-07-30

**Soundness:** 3 good
**Presentation:** 3 good
**Contribution:** 3 good
**Rating:** 6
**Confidence:** 3

**Summary:**

The paper studies new coordinate descent-type methods for equality-constrained problems, where 2 coordinates are updated on each iteration and proves new convergence guarantees under suitable proximal-PL conditions that allow to obtain linear convergence rates for the proposed methods.
The first main result considers the case of a single constraint on the sum of coordinates with a simple greedy rule for selecting the two coordinates to update. A linear dimension-independent convergence rate is established using a fixed step-size and assuming PL inequality w.r.t. 1 norm.

Next the authors consider the more challenging setting in which on top to the constraints on the sum of coordinates, there are also box constraints on the individual coordinates. Here the authors propose to use a greedy-proixmal update named Gauss-Southwell-q which leads to a linear rate using a fixed step-size and assuming PL inequality w.r.t. 2 norm, however with linear dependence in exponent of the dimension.

Finally, in the latter setting (equality constraint on sum + box constraints), considering a PL inequality w.r.t. 1 norm and a diffrerent greedy rule (which corresponds to solving a proximal-style problem w.r.t to 1 norm, which given the gradient direction could be solved in O(nlogn) time), the authors obtain a linear dimension-independent rate.

The proofs hinge on relating the 2-coordinate descent updates to the steepest descent method w.r.t. the 1 norm, which is interesting.

The authors present experiments that support their findings on a random least squares problem with equality (and then also box) constraints.



**Strengths:**

The paper is very well written and is easy to follow. The results and ideas are clearly presented. I enjoyed reading it.

The problems considered are sound and make sense, and the type of methods discussed is of clear practical interest to the community.

The technical argument of relating 2-coordinate descent to steepest descent w.r.t. 1 norm is interesting and might be of further use.

The convergence rates obtained are novel and interesting to the best of my knowledge.

**Weaknesses:**

1. I think the authors could do a better job in rigorously comparing the complexity of their methods to previous ones. It is not clear at all times if the obtained complexities are state-of-the-art (when not considering only 2-CD greedy methods). In particular, from first paragraph of 2nd page, it is not clear how this compete with known random selection methods.

2. To continue the above point, it seems all selection rules suggested require the full gradient direction. However, in such cases why these methods are better than full-gradient methods? For instance, for unconstrained least squares indeed doing an update to a single coordinate is much faster than computing entire gradient, and makes perfect sense when using random selection (e.g., w.r.t. to coordinate-wise Lipchitz parameters). But in the greedy rules we already need the full gradient.

3. The linear rates that are dimension-independent rely on PL w.r.t 1 norm. This might seem as ``hiding the dimension under the choice of norm''. Could the authors provide examples of interest that will satisfy this inequality with constant independent of the dimension? Or more precisely, that the ratio of the 1 norm constant and 2 norm constant is dimension independent?

4. At some point the authors claim that computing the coordinate-wise Lipchitz parameters can be difficult, however, in the most relevent setting I guess which allows also for the PL inequality: f(x) = g(Ax), with g strongly convex, in case g is well conditioned (e.g., squared Euclid norm in many cases), it is easy to estimate them I believe from the columns of A.

5. Throughout the paper the authors mention SVM as a major application, so why not conduct experiments with this application?

**Questions:**

See weaknesses.

---

> ### Author Rebuttal · Authors · 2023-08-09
>
> Thank you for highlighting these strengths. We put in a lot of work to ultimately find what we believe is a simple and elegant analysis of this issue; our older drafts of the paper had much more complicated analyses while achieving slower rates. We comment on the highlighted weaknesses below.
>
> > 1. "I think the authors could do a better job in rigorously comparing the complexity of their methods to previous ones."
>
> Can the reviewer be more specific about what is missing here?
>
> Several months before submission we contacted several authors of the works on random 2-CD methods (such as Patrascu) and asked them specifically “if you think we have accurately/fairly cited your work and other related works in the area” (we asked them to focus on what is now Section 2.4 in particular). Subject to minor suggestions that we incorporated before submission they did not have any objections.
>
> > 2. "in the greedy rules we already need the full gradient"
>
> While this is true, greedy rules do not need to compute the gradient from scratch at each step. There are many problems where we can track the full gradient for a lower cost than computing the full gradient. For example, for SVMs in the typical case of a dense kernel:
>
> - Computing the full gradient from scratch requires $O(n^2)$.
> - Computing a single element of the gradient requires $O(n)$.
> - Updating the full gradient after a single-coordinate update costs $O(n)$.
>
> So when we do coordinate descent in the above setting, we can track the full gradient for the same $O(n)$ cost required for random/cyclic selection. The above is the key to the efficiency of LIBSVM. There also exist other important problems where we can track the gradient for a similar cost to computing a random element of the gradient. See Nutini et al. [2015, Appendix A] for an in-depth discussion.
>
> > 3. "hiding the dimension under the choice of norm"
>
> Please see the discussion of dimension independence in the general reply, as well as the response to reviewer TqZB (for cases where $\mu_1$ and $\mu_2$ differ by much less than $n$).
>
> > 4. "coordinate-wise Lipchitz parameters ... it is easy to estimate them"
>
> For many typical problems where we apply coordinate descent it is indeed easy to estimate the coordinate-wise Lipshitz constants (e.g., squared 2-norm of columns for least squares). When discussing the challenge of using coordinate-wise Lipschitz constants, we are referring to the $O(n^2)$ cost of computing greedy rules that incorporate these constants (even without bound constraints). We will clarify this in the paper.
>
> > 5. "SVM as a major application"
>
> Please see the discussion of experiments in the general reply.

---

> > ### Comment · Reviewer_yjwM · 2023-08-18
> > **Response**
> >
> > I thank the authors for their comments.
> >
> > Regarding comparison to previous work, I think that usually a table does the best job for comparing complexity bounds of different methods, but I leave it to the authors to decide.
> >
> > Regarding other discussions, in particular concrete discussions where the methods presented can be superior to other, I would be happy if you include them in your revision.
> >
> > I am maintaining my score.

---

### Official Review · Reviewer_TqZB · 2023-07-30

**Soundness:** 2 fair
**Presentation:** 2 fair
**Contribution:** 2 fair
**Rating:** 5
**Confidence:** 3

**Summary:**

This work studies minimizing a smooth function with a summation equality constraint over its variables. The authors show a connection between the greedy 2-coordinate update and steepest descent w.r.t. 1-norm, and introduce a new proximal PL assumption w.r.t. 1-norm. They show improved convergence rates under such assumption over random selection. The authors also consider coordinate-wise Lipschitz smoothness and introduce an approximation for greedy 2-coordinate methods. They complement their theoretical results through numerical experiments on randomly generated problems.

**Strengths:**

1. The connection from 2-coordinate greedy update to steepest descent w.r.t. 1 norm seems non-trival and potentially interesting for equality-constrained problems.
2. The authors offer detailed comparisons on various Gauss-Southwell update rules (GS-1, GS-q and GS-s).

**Weaknesses:**

I believe this work has potential but lacks a few key ingredients.
1) This work lacks justification for their proximal-PL condition w.r.t. 1-norm. For example, Karimi et al (2016) provides five important classes of functions that statisfy the proximal-PL condition w.r.t. 2-norm. It is unclear to me what function classes can statisfy the proximal-PL condition w.r.t. 1-norm where the worst case dependence on $n$ for $\mu_1$ can be avoided. It would also be useful if there can be empirical comparisions for $\mu_1$ against $\mu_2$ as $n$ scales. At its current state, I do not think it is reasonable to claim that the convergence rate is independent of the problem dimension $n$.
2) I would like to see experiments performed on practical datasets (e.g., LIBSVM datasets) rather than randomly generated Gaussian data, since much of the theoretical results aim to improve upon the rules in LIBSVM. I would also like to see experiments on SVM problems to support their claimed improvements for SVM.
3) I suggest adding markers to the plots in Figure 1 and 2, as the overlapping nature of the results renders some plots unparsable.

**Questions:**

1. What are the main differences of GS-1 rule (Algorithm 1) for proximal-PL assumption w.r.t. 1-norm, compared to Song et al. (2017)?

**Limitations:**

N/A.

---

> ### Author Rebuttal · Authors · 2023-08-09
>
> Thank you for the suggestions on improving the plots.
>
> > "It is unclear to me what function classes can satisfy the proximal-PL condition w.r.t. 1-norm where the worst case dependence on $n$ for $\mu_1$ can be avoided."
>
> It is a good point that the paper currently does not give an example where $\mu_1$ is better than $\mu_2/n$. We will add a reference to Sections 4.1 and 4.3 of Nutini et al. [2015], who give examples where $\mu_1$ becomes arbitrarily close to $\mu_2$ as $n$ grows. Since these examples are for unconstrained, strongly-convex minimization, we have extended Appendix C to show that strong-convexity of $f$ in the 1-norm implies proximal-PL in the 1-norm with the same constant (i.e. $\mu_1$). Thus, the examples of Nutini et al. apply directly to our constrained setting.
>
> Unfortunately, it is not feasible to compare $\mu_1$ to $\mu_2$ numerically for complicated problems since it is not known how to compute $\mu_1$ explicitly except in simple cases.
>
> > "At its current state, I do not think it is reasonable to claim that the convergence rate is independent of the problem dimension."
>
> Please see the discussion of dimension-independence in the general reply.
>
> > “I would like to see experiments performed on practical datasets… I would also like to see experiments on SVM problems to support their claimed improvements for SVM.”
>
> Please see the discussion of experiments in the general response.
>
> > “What are the main differences of GS-1 rule (Algorithm 1) for proximal-PL assumption w.r.t. 1-norm, compared to Song et al. (2017)?”
>
> While this work and Song et al. both consider using steepest descent in the 1-norm, the problem settings are different: Song et al. consider L1-regularization where you may only need to update 1 variable, while we consider bound and summation constraints which is more complicated because you must always update at least 2 variables. So, the algebraic definitions of the GS-1 rule are the same, but the problem-specific implementations are very different.

---

> > ### Comment · Reviewer_TqZB · 2023-08-16
> > **Response to Authors' Rebuttal**
> >
> > I thank the authors for the replies and am glad to know that they find my suggestions helpful. My concerns regarding $\mu_1$ vs $\mu_2$ and the comparison to Song et al. (2017) have been properly addressed.
> >
> > However, I maintain my reserve regarding their subpar experimental results for a ML venue. While this is mostly a theoretical work, it concerns a practical problem with well-defined experimental settings using real datasets. I am not fully convinced by the results in the attached PDF, although I do understand that it is already a big ask for the authors to add these results within such short timeframe, and I thank the authors for that.
> >
> > As such, I am increasing my score.

---

> > > ### Author Response · Authors · 2023-08-17
> > >
> > > Thank you for reading and considering the response. We plan to add a comprehensive set of experiments on real data (beyond what is included in the author response).

---

### Author Rebuttal · Authors · 2023-08-09

We thank the reviewers for taking the time to read the paper and provide feedback on our work. We believe that the paper will be strengthened by incorporating this input. Below we comment on two issues that were brought up in multiple reviews.

**Experiments on real data**

We view our primary contribution as providing the first theoretical proof of the fast convergence of greedy 2-coordinate methods. While the submitted version did not perform experiments on real data, we note that the greedy 2-coordinate code LIBSVM has around 50,000 citations, indicating that practitioners find this approach useful in a variety of practical scenarios. We view providing a theoretical grounding for this approach as the most important contribution of our work and thought further experiments on this topic would be redundant.

Nevertheless, it is true that there are new methods proposed in the paper and it would be interesting to see their performance on real datasets. We are thus now adding experiments on real datasets to the paper. We have found that the performance on real data seems to be similar to the performance on simulated data, although in some cases the GS-1 rule performed better on real data than we expected. As examples, we include experiments on 3 datasets from the LIBSVM webpage in the attached PDF.

**Dimension independence**

In this work we use "dimension independence" as it is used in the optimization literature. Namely, a rate is dimension independent if changing the dimension of the problem (but not other constants) does not change the convergence rate.

For example, the gradient descent rate $(1-\mu_2/L)$ is dimension independent since if we change the dimension $n$ but keep $\mu_2$ and $L$ constant then it converges at the same speed. This is true even though we might expect $L$ to be larger and $\mu_2$ to be smaller for high-dimensional problems. This is in contrast to a random coordinate descent rate of $(1-\mu_2/nL_c)$ which is dimension dependent: it becomes slower as the dimension $n$ increases even with fixed $\mu_2$ and $L_c$ (the coordinate-wise Lipschitz constant of the gradient). Our $(1-2\mu_1/L_2)$ rate is dimension independent since the rate does not change if $n$ changes but $\mu_1$ and $L_2$ are fixed. Notably, our greedy coordinate descent rates are meaningful even if we consider infinite dimensional problems. This is similar to gradient descent.

---

### Decision · Program_Chairs · 2023-09-21

**Decision:**

Reject

**Comment:**

After aggregating all information from reviewers, authors feedback, and going over the paper myself, this was one of the borderline papers for which making the decision was the hardest. While the reviewers have certainly appreciated the paper and the results, at the bottom line, and after carefully considering all data, it is my decision that this paper is not yet ready for publication. On the theoretical part, certain aspects of the writing and discussion needs to be improved including the different choice of norms for the PL inequality and why it makes sense (in a significant way, beyond toy examples), crystal-clear comparison of complexity bounds to related methods, and relevant examples for which the method has SOTA complexity (rate and runtime), all these aspects could be improved and I urge the authors to take the feedback into consideration. On the experimental side, a few reviewers (three of them) where not content with the numerical results, even post feedback, and the authors themselves have mentioned their aim of extensively improving the experiments. Thus, overall, I think this paper would benefit from a carful revision and resubmission.